# Simulation of storm surge inundation under different typhoon intensity scenarios: Case study of Pingyang County, China

Xianwu Shi[1,2,4]• Pubing Yu[3]• Zhixing Guo[1]• Zhilin Sun[2]• Fuyuan Chen[2]• Xiuguang Wu[2]• Wenlong Cheng[2]•Jian Zeng[2]

[1]National Marine Hazard Mitigation Service, Beijing, 100194, China
[2]Zhejiang University, Hangzhou, 310058, China
[3]Zhejiang Institute of Hydraulics and Estuary, Hangzhou, 310020, China
[4]Key Laboratory of Ministry of Education for Coastal Disaster and Protection, Hohai University, Nanjing 210098, China
*Correspondence to*: Jian Zeng(zengjian@zjwater.gov.cn)

**Abstract**: China is one of the countries that are most seriously affected by storm surges. In recent years, storm surges in coastal areas of China have caused huge economic losses and a large number of human casualties. Knowledge of the inundation range and water depth of storm surges under different typhoon intensities could assist pre-disaster risk assessment and making evacuation plans, as well as provide decision support for responding to storm surges. Taking Pingyang County in Zhejiang Province as a case study area, parameters including typhoon tracks, radius of maximum wind speed, astronomical tide, and upstream flood runoff were determined for different typhoon intensities. Numerical simulations were conducted using these parameters to investigate the inundation range and water depth distribution of storm surges in Pingyang County considering the impact of seawall collapse under five different intensity scenarios (corresponding to minimum central pressure values equal to 915, 925, 935, 945, and 965 hPa) . The inundated area ranged from 103.51 km$^2$ to 233.16 km$^2$ for the most intense typhoon. The proposed method could be easily adopted in various coastal counties and serves as an effective tool for the decision making in storm surge disaster risk reduction practices.

**Keywords**: Intensity scenarios; Inundation simulation; Typhoon-induced storm surge; Pingyang County;

**1 Introduction**

China is among the few countries affected seriously by storm surges. A storm surge can cause overflow of tide water and seawall destruction that can result in flooding in coastal areas, which can be extremely destructive and can have serious impact on surrounding areas (Sun et al. 2015). Storm surges have occurred along much of China's coast from south to north (Gao et al. 2014). On average, approximately nine typhoons annually make landfall over China (Shi et al. 2015), most of which cause storm surges. In 2018, storm surge disasters caused coastal flooding in China that resulted in 3 deaths and direct economic losses of RMB 4.456 billion (Ministry of Natural Resources 2019). With the recent rapid socioeconomic development in China, industrialization and urbanization processes in coastal areas have accelerated, and both the population density and the social wealth in such areas have increased sharply. Concurrently, owing to global climate change and sea level rise, the occurrence of weather situations that trigger storm surges has become more frequent and the associated risk level of coastal storm surges has increased significantly (Fang et al., 2014; Yasser et al., 2018). Fortunately, the number of fatalities in China due to storm surges has decreased significantly because of improvements in the regional early warning capability (Shi et al. 2015). Thus, the focus on storm surge disasters has changed from reduction of disaster losses to mitigation of disaster risks. Therefore, research on storm surge risk has been attracted more and more attention (Shi et al., 2019).

Storm surge risk assessment aims to estimate the risk level of storm surges in a certain region based on deterministic numerical simulation in combination with designed probabilistic storm surge scenarios (Shi et al. 2013; Wang et al. 2018). The calculation of storm surge under scenarios with storms of different intensity is an important part of storm surge risk assessment. The calculation results could provide important decision-making support for the response to storm surges in coastal areas, and they could assist in both the pre-assessment of storm surge disasters and the preparation of storm surge emergency evacuation plans. Following the earthquake-induced "3.11" tsunami that occurred in Japan in 2011, scientific research on many aspects of marine disaster risk management became of great concern to various governments. With consideration of storm surge disaster as the primary hazard, China commenced a project for marine disaster risk assessment and zoning, and it subsequently released its marine industry standard, *the Technical Guidelines for Risk Assessment and Zoning of Marine Disaster Part 1: Storm Surge* (Liu et al. 2018). Calculation of the inundation range and water depth of storm surges associated with typhoons of different intensity is one of the most important tasks in storm surge risk assessment.

The core element of simulation of inundation by storm surge disaster under scenarios of different typhoon intensity is to set key parameters for both the typhoons and the storm tides (e.g., typhoon track, typhoon intensity, radius of maximum wind speed, and astronomical tide) under different conditions (Shi et al, 2020). Tomohiro et al. (2010) set key parameters for the largest possible typhoon-induced storm surge in different regions of Japan by simulating typhoon track translation using indicators of the Ise Bay typhoon (the most serious typhoon event recorded in Japan's history) as reference typhoon parameters. To overcome the limitation of historical records , a stochastic modelling method has been developed for simulation of typhoon track and intensity. This method is to analyze the statistical probability characteristics of historical typhoons in terms of their annual

frequency, seasonal distribution, track distribution, intensity, and influence areas. Based on these features, the generation, development, and lysis of typhoons can be simulated to generate a large number of typhoon events (Powell et al., 2005; Lin et al., 2010). By selecting events with different

typhoon intensity from the generated samples, the inundation extents and depths of the study area can be calculated using the storm surge numerical model (Wood et al. 2006; Wahl et al. 2015), and these researches mainly focus on the coast of North Atlantic Ocean. Considering the typhoon landing and historical storm surge events happened in the coastal areas of China, how to set the parameters for performing the simulation of typhoon-induced storm surge under different typhoon

intensity scenarios is an interesting and important topic towards the coast of China.

This study considered Pingyang County of Zhejiang Province (China) as a case study area. The objective was to propose a deterministic method to calculate the inundation extents and depths caused by different typhoon intensity scenarios combined with the storm surge numerical model. The key parameters (e.g., typhoon intensity, typhoon track, maximum wind speed radius) corresponding to the characteristics of

typhoons landing the coastal areas of China was set. The astronomical tide, upstream flood runoff and seawall collapse was taken into consideration as an important factor in the storm surge simulation. The results aim to contribute to the methodology of quantitative assessment of storm surge hazards for coastal counties.

**2 Materials**

**2.1 Case study area**

Pingyang County is a coastal county belonging to the city of Wenzhou in Zhejiang Province, China (Fig. 1) and is affected most frequently by storm surge in coastal areas. It is located in the tropical storm zone of the western Pacific Ocean and is generally exposed to the risk of storm surges during July–October. Pingyang County lies within the region 27°21′–27°46′N, 120°24′–121°08′E, and it is bordered by Ruian,

Wencheng, Taishun, and Cangnan counties. The county extends roughly 83 km from east to west and roughly 25.4 km from north to south, covering an area of approximately 1051 km2. It is a highly developed and densely populated area of China with considerable asset exposure. Pingyang County has a population of approximately 800,000, and it is the first National Coastal Economic Open County with customs, ports, and important industries. The coastal zone of Pingyang County, which extends for 22 km,

is surrounded by sea on its eastern, southern, and northern sides. The Ao Jiang and Feiyun Jiang flow across the county and they discharge into the East China Sea. Storm surges frequently hit this county, which is one of the reasons why the China State Oceanic Administration approved Pingyang County as the first National Marine Disaster Mitigation Comprehensive Demonstration Area in China.

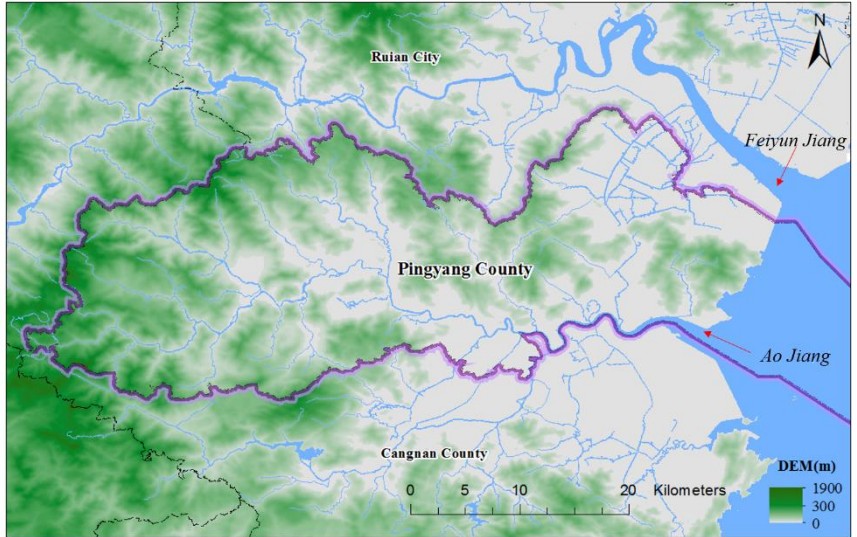

**Fig. 1 Case study area**

### 2.2 Data

Multisource data (Table 1) were collected to perform a storm surge numerical modelling in Pingyang county. The digital elevation map (DEM) of Pingyang county and nearshore submarine topography data were collected to construct the numerical model, and tidal observational data were used to validate the model. Historical typhoon records, including time, location, and intensity, were collected to set the typhoon parameters driving the storm surge numerical model. A field survey was carried out by Zhejiang Institute of Hydraulics and Estuary to investigate the inundation situation along the Ao Jiang river in Pingyang County. Researchers equipped with GPS-RTK (Global Positioning Systems, Real-Time Kinematic) and rangefinders worked in two groups to make measurements from Oct.2nd to Oct.7th in 2013. The extent of the inundation was estimated based on flooding marks, such as the accumulation of trash, signs of mud, and withered plants. In addition, the extent of inundation was established through interviews with local residents.

**Table 1 Multisource data used to perform storm surge numerical modelling in Pingyang County**

| Data type | Time series | Description | Source |
|---|---|---|---|
| Historical typhoon records | 1949–2018 | Time, location, and intensity of each typhoon track point | Shanghai Typhoon Institute, China Meteorological Administration |
| Digital elevation map and submarine topography | 2015 | Digital elevation map of Pingyang County and offshore submarine topography | Surveying and Mapping Bureau of Zhejiang Province |
| Tidal observational data | 1990–2015 | Hourly observational data of surge and water level for tidal station during typhoon periods | East China Sea Marine Forecasting Center, Oceanic Administration of China |
| Historical inundation ranges | 2013 | Inundation ranges caused by Fitow along the Aojiang river in Pingyang County | Field surveying by Zhejiang Institute of Hydraulics and Estuary |

### 3 Methods

This study proposed a framework for calculation of storm surge inundation simulation under different typhoon intensity scenarios (Fig.2). The proposed framework was composed by four parts: model configuration, model validation, parameters setting and inundation simulation. The numerical model was used to simulate the storm surge inundated range and water depth, and the DEM and nearshore submarine

topography data was used to construct the storm surge numerical model. The numerical model was validated by historical observational data of tidal station and field-surveying data of inundated areas. Based on the historical observational data, the key parameters (e.g., typhoon track, radius of maximum wind speed, astronomical tide, and upstream flood runoff) could be set to drive the storm surge numerical model. The proposed method could be easily adopted in various coastal counties and serves as an effective tool for the decision making in storm surge disaster risk reduction practices.

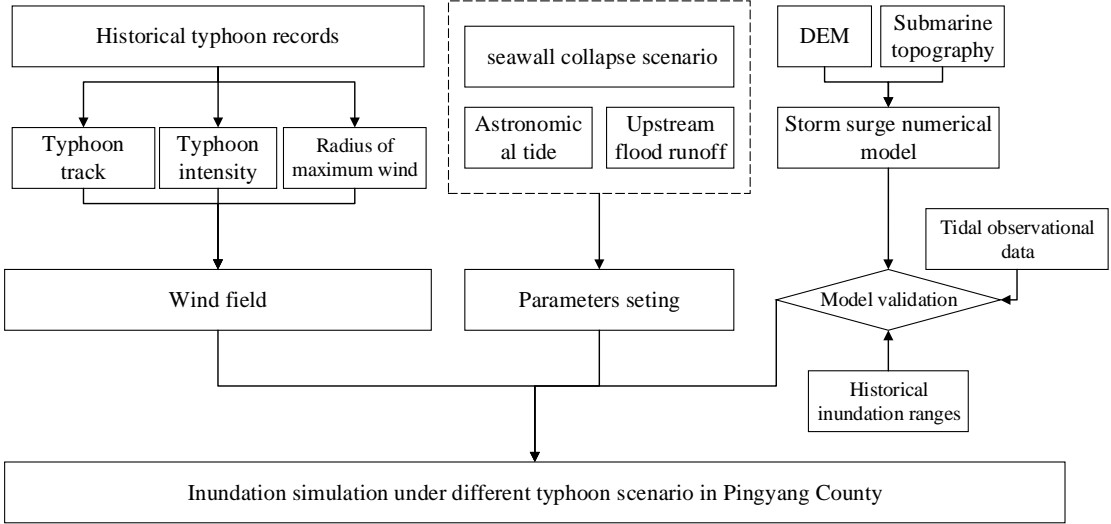

**Fig. 2 Framework of storm surge inundation simulation under diffirent typhoon scenario in Pingyang County**

### 3.1 Model configuration

The typhoon-astronomical-flood-wave coupled numerical model used in this study, developed by the Zhejiang Institute of Hydraulics and Estuary, is based on the unstructured-grid finite volume method, and more detailed model information could be found in Chen et al (2019). It has characteristics of high efficiency, accuracy, conservation, and automatic capture of intermittent flow. This model could be used to simulate conventional river channel flow and offshore water flow including flood evolution, astronomical tides, storm surges, and flooding. The storm surge simulations were performed in combination with the calculation of river runoff and consideration of typhoon wind and air pressure fields. Thus, the model was capable of simulating large-scale detailed storm surge flooding. The open-sea boundary of the model was set near the first typhoon warning line in China's offshore area (Fig. 3a). The model contained triangular and quadrilateral grids consisting of 258,543 units and 173,910 nodes. The coverage extended to Bohai Bay and the Sea of Japan in the north, to the south of Taiwan in the south, and to the east of the Ryukyu Islands (Fig. 3a) in the east. The upper boundary of the Aojiang River was set at Daitou, the upper boundary of the Feiyun River was at Zhaoshandu, and the upper boundary of the Oujiang River was set at Weiren. The grid for the offshore area and land in Pingyang County with elevation of <30 m (including Nanji Island) had a side length of 50–200 m, and the minimum side length was 15 m (Fig. 3b). The model covered the Bohai Sea, Yellow Sea, East China Sea, Sea of Japan, Korean Strait, Taiwan Strait, Yangtze River Estuary, Hangzhou Bay, and Qiantang River. The mesh grid was finest in localized areas of the Zhejiang offshore region, Oujiang River Estuary, Feiyun River Estuary, and Aojiang River Estuary.

Waves caused by typhoon are simulated by Simulating Waves Nearshore (SWAN) model in this study

(Booij et al, 1999). The SWAN model is a third-generation numerical wave model, which is used to

simulate wind-generated wave propagation in coastal regions. It can describe the evolution of wave fields

under specific wind, flow and underwater terrain conditions in shallow waters. The governing equation

is as follows.

$$\frac{\partial}{\partial_t}N + \frac{\partial}{\partial_x}C_x N + \frac{\partial}{\partial_y}C_y N + \frac{\partial}{\partial_\sigma}C_\sigma N + \frac{\partial}{\partial_\theta}C_\theta N = \frac{S}{\sigma} \tag{1}$$

In the equation, N is wave action, $\sigma$ is relative frequency of waves, $\theta$ is wave direction, and S is source

item. $C_x$ and $C_y$ are wave propagation speed in $x$ and $y$ direction, respectively. $C_\sigma$ is propagation speed

of wave action in frequency space, and $C_\theta$ is the propagation speed of wave action in wave direction

space. Based on the wave elements and the structural parameters of the seawall, the overtopping

discharge is calculated by the empirical formula. In the simulation of dike-breaching, the varying dike

top elevation is applied according to overtopping discharge to simulate the process of dike-breaching.

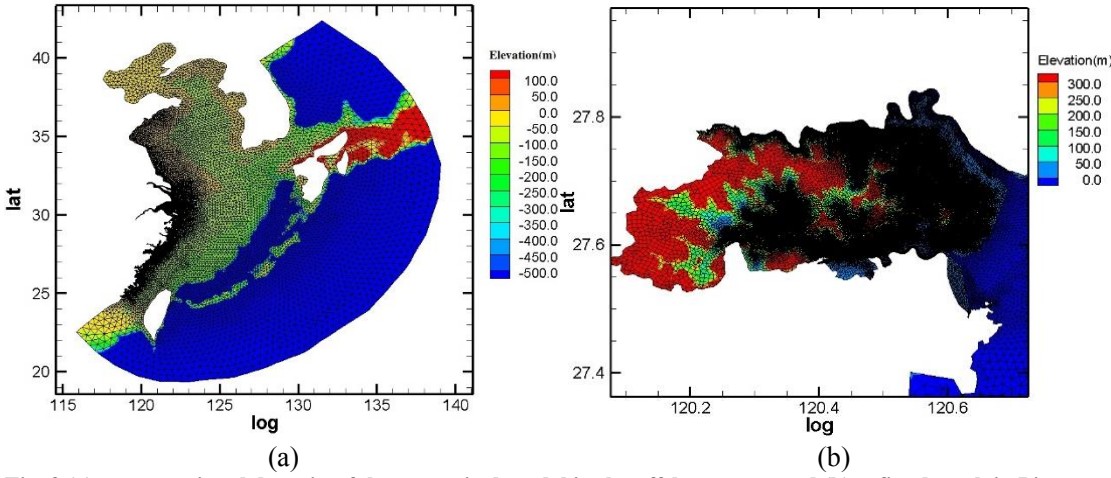

(a)                                    (b)

**Fig. 3 (a) computational domain of the numerical model in the offshore area, and (b) refined mesh in Pingyang County**

**3.2 External force**

The storm surge numerical model was driven by wind stress and the atmospheric pressure gradient acting

on the surface. The Jelesnianski model was chosen to generate the wind and pressure fields (Jelesnianski,

1965), for which the calculation formulas are as follows:

$$W = \begin{cases} \frac{r}{r+R}\left(V_{0x}\vec{\imath} + V_{0y}\vec{\jmath}\right) + W_R\left(\frac{r}{R}\right)^{\frac{3}{2}}\frac{(A\vec{\imath}+B\vec{\jmath})}{r}, & (0<r\leq R) \\ \frac{R}{r+R}\left(V_{0x}\vec{\imath} + V_{0y}\vec{\jmath}\right) + W_R\left(\frac{R}{r}\right)^{\beta}\frac{(A\vec{\imath}+B\vec{\jmath})}{r}, & (r>R) \end{cases} \tag{2}$$

$$P_a = \begin{cases} P_0 + \frac{1}{4}(P_\infty - P_0)\left(\frac{r}{R}\right)^3, & (0 < r \leq R) \\ P_\infty - \frac{3}{4}(P_\infty - P_0)\frac{R}{r}, & (r > R) \end{cases} \tag{3}$$

and

$$A = -[(x - x_c)\sin\theta + (y - y_c)\cos\theta] \tag{4}$$
$$B = (x - x_c)\cos\theta + (y - y_c)\sin\theta \tag{5}$$

In the above equations, R is the radius of maximum wind speed, r is the distance from the calculated

point to the center of the typhoon, $(V_{0x},\ V_{0y})$ is the translation speed of the typhoon, $(x,\ y)$ and

$(x_c,\ y_c)$ are the coordinates of the calculated point and the typhoon center, respectively, $\theta$ is the inflow

angle, $P_0$ is the central pressure of the typhoon, $P_\infty$ is the atmospheric pressure at infinite distance, and

$W_R$ is the maximum wind speed of the typhoon.

**3.3 Model verification**

Verification of the storm surge numerical model was performed using 20 typhoon-induced storm surge events that affected Pingyang County during 1990 to 2015. The tidal range along Pingyang coast is about 4.45m. The differences between the simulated and observed water level and surge of the 20 typhoon-induced storm surge events were compiled at six tidal stations (Jiantiao, Haimen, Kanmen, Dongtou, Ruian, and Minjiang as shown in Fig.4 ) close to Pingyang County as shown in Table 2 and 3. According

to the statistical results, the absolute average error of the water level of all involved tidal stations was 12–18 cm, and the absolute average maximum storm surge error of all involved tidal stations was 11–15 cm. Storm surge caused by typhoon Fitow (No. 1323) is the most serious event that affected Pingyang County in the past ten years. Verification of water level and storm surge of 6 tidal stations affected by typhoon Fitow (No. 1323) is presented in Fig. 5 and 6, respectively. It can be seen that both the phase

and the water level obtained from the storm surge simulation are highly consistent with the actual measurements, proving that the storm surge numerical model developed in this study was reliable.

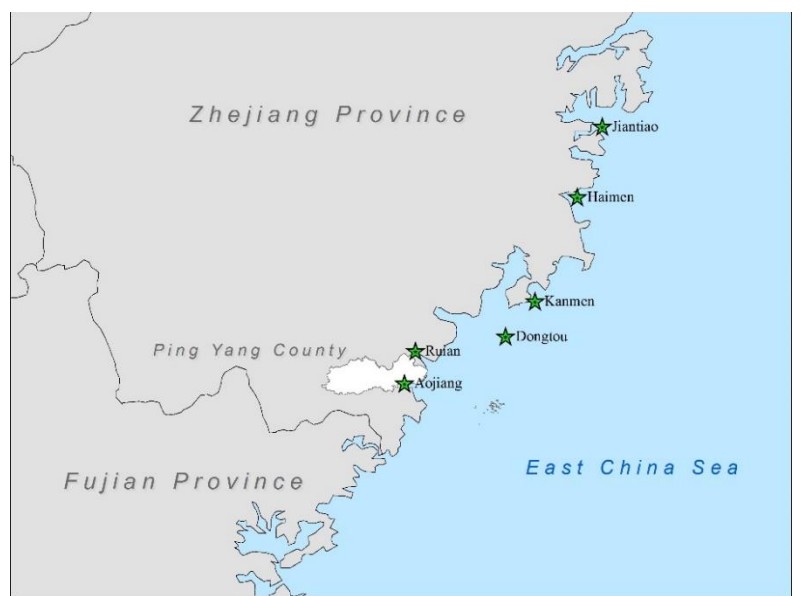

**Fig.4 the distribution of tidal station along the coastal Zhejing Province**


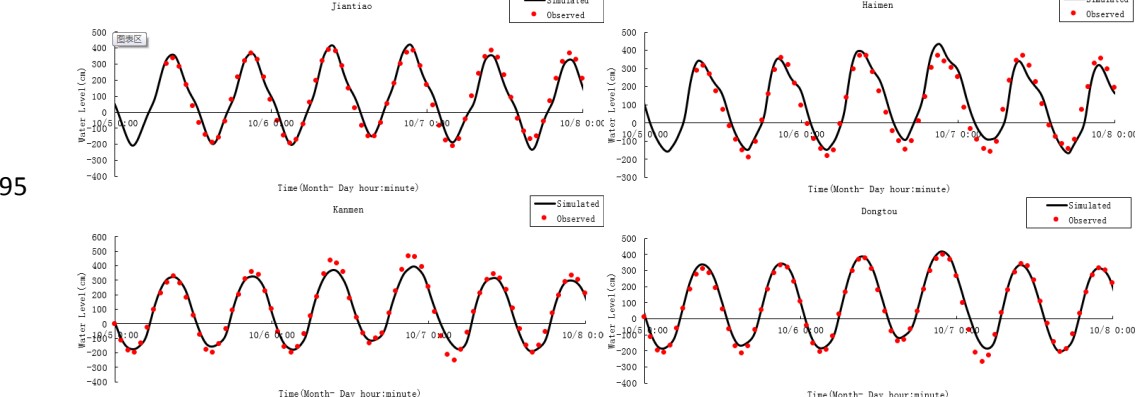

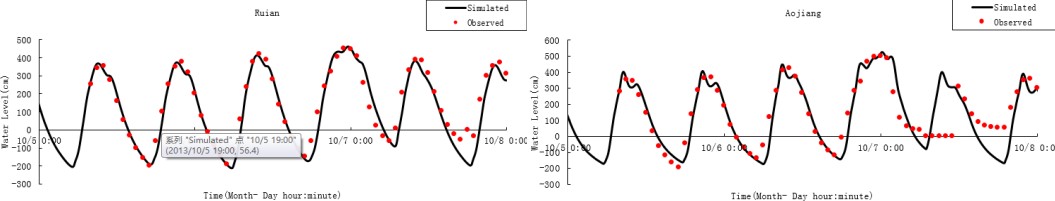

**Fig. 5 Verification of the high tide level for tidal stations affected by the storm surge caused by Typhoon Fitow (No. 1323)**

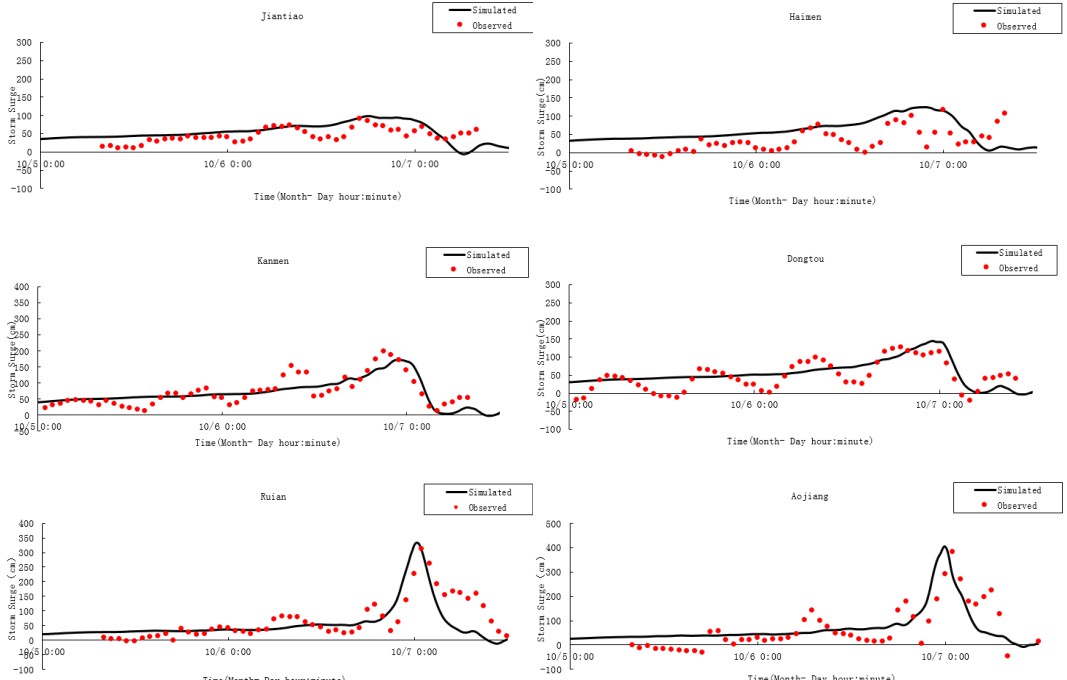

**Fig. 6 Verification of the storm surge for tidal stations affected by the storm surge caused by Typhoon Fitow (No. 1323)**

**Table 2 Error statistics in terms of maximum water level during landfall of 20 typical typhoons (unit: cm)**

|      | Jiantiao | Haimen | Kanmen | Dongtou | Ruian | Aojiang | Average |
|------|----------|--------|--------|---------|-------|---------|---------|
| 9015 | —— | —— | 3 | 2 | -23 | —— | 9 |
| 9216 | -6 | -24 | —— | -15 | -1 | 4 | 10 |
| 9219 | 20 | 23 | -1 | 18 | -7 | -3 | 12 |
| 9417 | 14 | 4 | —— | 31 | 11 | -3 | 11 |
| 9608 | -19 | -20 | —— | -30 | -25 | 2 | 19 |
| 9711 | -17 | 11 | -24 | —— | -24 | -31 | 22 |
| 0004 | 16 | 25 | -20 | 1 | 14 | 15 | 15 |
| 0008 | 12 | -1 | -26 | 12 | 7 | 13 | 12 |
| 0108 | 18 | 14 | 16 | 17 | 24 | 22 | 19 |
| 0216 | 5 | -16 | -11 | 17 | 21 | 17 | 15 |
| 0505 | 31 | -5 | -31 | -33 | -15 | 31 | 24 |
| 0509 | -17 | -17 | -14 | -14 | 2 | 7 | 12 |
| 0515 | 0 | 28 | -17 | -4 | 1 | 6 | 9 |
| 0604 | -2 | -14 | -24 | -9 | -17 | -1 | 11 |
| 0608 | 1 | -13 | -2 | 3 | 15 | 23 | 10 |
| 0713 | 16 | 4 | 10 | -1 | 25 | 26 | 14 |
| 0716 | 19 | 8 | 14 | 13 | 21 | 15 | 15 |

| | | | | | | | |
|---|---|---|---|---|---|---|---|
| 0908 | 20 | 22 | —— | —— | 10 | 2 | 14 |
| 1209 | 6 | -10 | -21 | -10 | 16 | 15 | 13 |
| 1323 | 26 | 31 | -55 | 19 | -8 | -17 | 26 |
| Average | 14 | 15 | 18 | 14 | 14 | 13 | 15 |

Note: "——"means no observational value.

**Table 3 Error statistics in terms of the maximum storm surge during landfall of 20 typical typhoons (unit: cm)**

| | Jiantiao | Haimen | Kanmen | Dongtou | Ruian | Aojiang | Average |
|---|---|---|---|---|---|---|---|
| 9015 | —— | —— | -11 | -17 | -9 | —— | 12 |
| 9216 | -8 | -6 | -34 | -6 | -2 | -12 | 11 |
| 9219 | 6 | 15 | -11 | 14 | 13 | 20 | 13 |
| 9417 | -21 | -17 | -2 | —— | -20 | -8 | 14 |
| 9608 | -10 | -21 | -35 | -10 | -16 | -10 | 17 |
| 9711 | 23 | 21 | -15 | -20 | -1 | -7 | 15 |
| 0004 | 10 | 12 | -14 | 7 | 10 | -4 | 10 |
| 0008 | 3 | -2 | -2 | 16 | -2 | 10 | 6 |
| 0108 | 10 | 10 | 1 | 27 | 5 | 3 | 9 |
| 0216 | 11 | 2 | 2 | 2 | 26 | 19 | 10 |
| 0505 | -15 | -5 | -26 | -10 | -26 | -38 | 20 |
| 0509 | -8 | 17 | -14 | 8 | -15 | -10 | 12 |
| 0515 | 9 | 12 | -20 | -12 | -23 | -10 | 14 |
| 0604 | -10 | -21 | -35 | 10 | 15 | 9 | 17 |
| 0608 | -1 | 5 | 10 | -13 | 13 | -3 | 8 |
| 0713 | 21 | -2 | -8 | 28 | 17 | 18 | 16 |
| 0716 | 9 | 15 | 20 | 2 | 17 | 21 | 14 |
| 0908 | -42 | -3 | —— | —— | -6 | -5 | 14 |
| 1209 | —— | —— | -6 | —— | 10 | 13 | 10 |
| 1323 | 5 | 7 | -27 | 12 | 21 | 19 | 15 |
| Average | 12 | 11 | 15 | 13 | 13 | 13 | 13 |

Note: "——" means no observational value.

Besides, a validation for the inundation simulation was performed based on the inundation ranges through field surveying. The model described above was used to perform a simulation of the area along the Aojiang river (Pingyang County) inundated by Typhoon Fitow. A field survey was undertaken by the Zhejiang Institute of Hydraulics and Estuary to investigate the inundation areas in Pingyang County during the storm surge disaster period caused by Fitow (Fig. 7b). The simulated and investigated inundation areas were compared (Fig. 7). It can be seen that the surveyed and simulated inundated areas are similar. The extent of the surveyed inundated area was slightly larger than that simulated because typhoon precipitation during the period of influence of Fitow caused urban waterlogging in parts of Pingyang County.

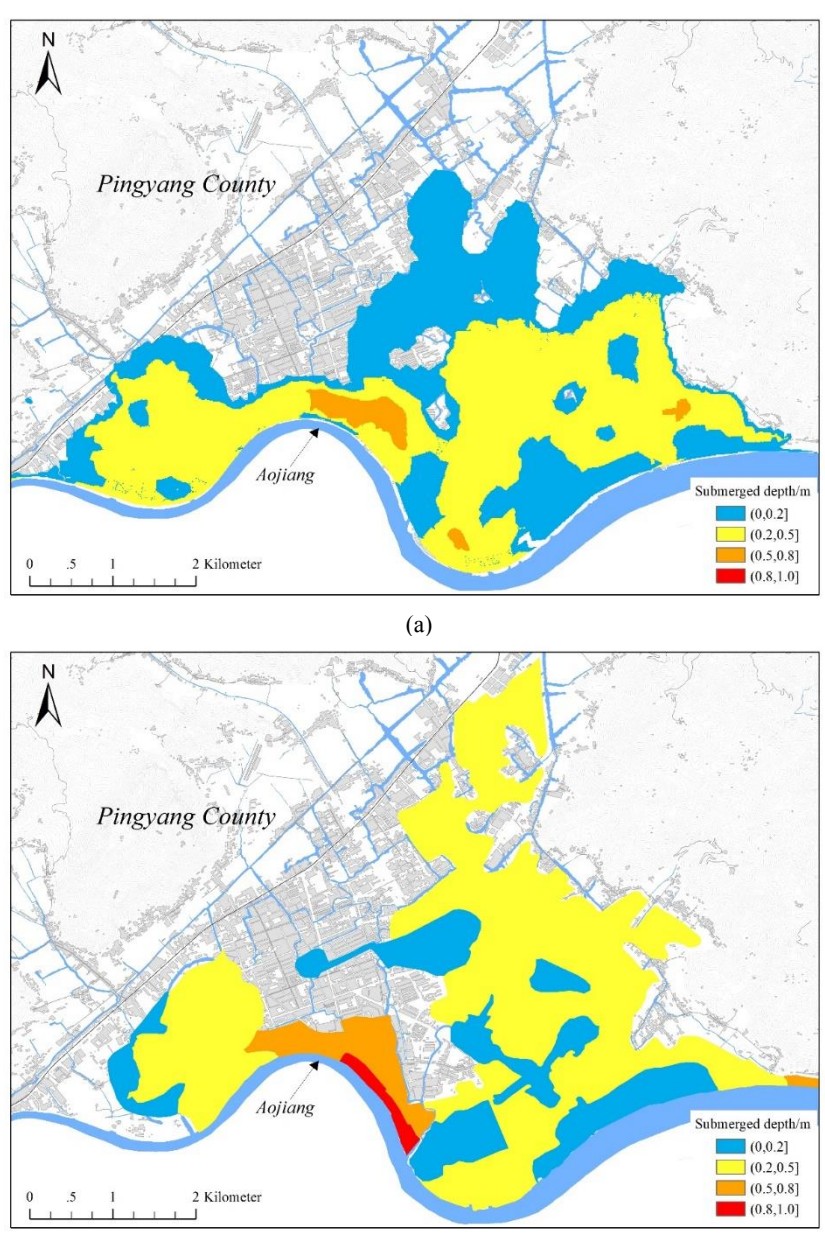

(a)

(b)

**Fig. 7 (a) Simulated inundated area and (b) surveyed inundated area during typhoon Fitow**

**3.4 Parameter setting**

**3.4.1 Typhoon intensity**

Pingyang County is frequently affected by typhoons, and is experienced 132 hazardous typhoons during 1951-2013 with an average of 2.13 times per year. The most intense recorded landing typhoon since 1950 that affected Pingyang County is Saomai (No. 0608), and the central pressure at the time of landing reaches 920 hPa with the wind speed of 60 m/s. About 40 percent of the typhoons at the time of landing with the central air pressure were lower than 965 hPa. The county has been affected by 20 typhoons with central air pressure in the range of 920 to985 hPa (average: 965 hPa) since 1990. Based on the actual needs for response to coastal storm surges, this study considered typhoons with five different levels of intensity (Table 4), which were based on the central air pressure during landfall with reference to *the Technical Guidelines for Risk Assessment and Zoning of Marine Disaster Part 1: Storm Surge* (Liu et al. 2018).

**Table 4 Typhoon intensity scenarios**

| Typhoon Intensity | I | II | III | IV | V |
|---|---|---|---|---|---|
| Maximum wind force | Level 12-13 | Level 14-15 | Level 16 | Level 17 | Above 17 |
| Minimum air central pressure (hPa) | 965 | 945 | 935 | 925 | 915 |

**3.4.2 Typhoon track**

This study selected the two typhoons that had the most severe impact on Pingyang County and that generated the most significant storm surge in history, i.e., Typhoon Fred (No. 9417) and Typhoon Saomai (No. 0608) as shown in Fig 8. Typhoon Fred caused the most severe storm surge in central and southern parts of Zhejiang Province (including Pingyang County) since 1949. The minimum central air pressure of this typhoon was 935 hPa; however, when making landfall near Ruian, the central air pressure was 960 hPa and the radius of maximum wind speed was approximately 50 km. This typhoon landed at the time of the highest astronomical tide and it generated the highest water level ever recorded in the coastal area, causing the water level at Ao Jiang Station up to 6.56m. Typhoon Saomai had the lowest central air pressure and the fastest wind speed of any typhoon since 1949. The minimum central air pressure reached 915 hPa; however, when making landfall near the Cangnan tidal station, the central air pressure was 920 hPa and the radius of maximum wind speed was approximately 15 km. Before making landfall, both typhoons traveled in a direction perpendicular to the shoreline, conducive to generating the greatest storm surge.

To determine which of these two typhoons had the track that caused the larger storm surge in Pingyang County under the same conditions, both were assumed to have central air pressure of 915 hPa, radius of maximum wind speed of 36 km, and constant direction of movement. The track of Typhoon Fred was translated to the landing site of Typhoon Saomai. The results showed that the maximum storm surge of Typhoon Fred and Typhoon Saomai was 7.22 and 7.47 m, respectively, at Aojiang and 7.00 and 7.03 m, respectively, at Ruian. As the storm surge associated with Typhoon Saomai was slightly larger, the track of this typhoon was selected as the designed typhoon track. The designed typhoon track was translated to a position in the middle of Pingyang County and then translated to the sides by a distance of 0.25 times the radius of maximum wind speed, until the track combination that maximized the storm surge in each coastal area of Pingyang County was determined (Fig.9). This track was then used for the inundation superposition calculation.

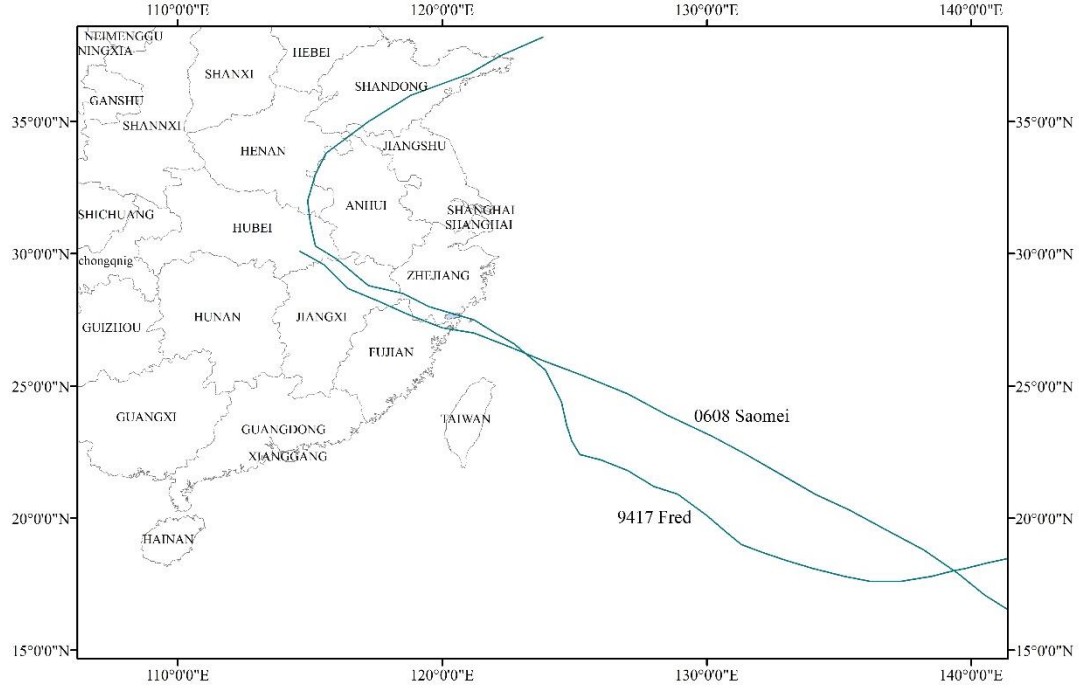

**Fig 8 Typhoon track of 9417 "Fred" and 0608"Saomei"**

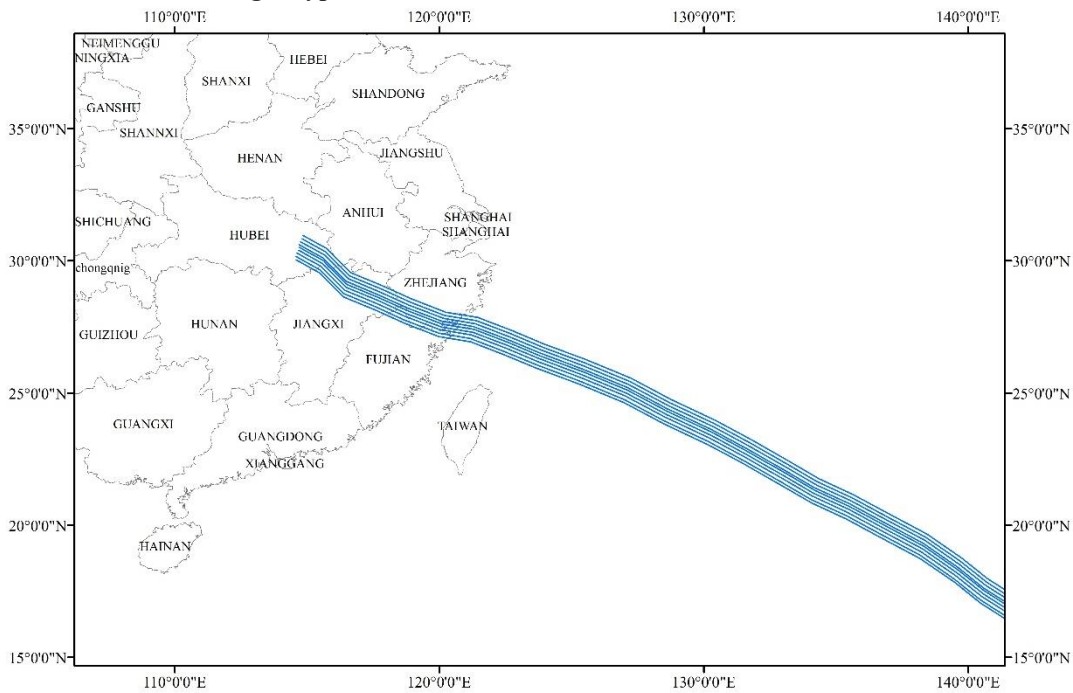

**Fig. 9 "designed typhoon" track set over Pingyang County**

### 3.4.3 Radius of maximum wind speed

The radius of maximum wind speed which is the radius from the typhoon's center to the position where the maximum wind speed occurs was used to indicate the "size" of the typhoon. The radius of maximum wind speed of a typhoon is an important factor for the simulating of storm surge height and coastal inundation extent. Collecting the historical radius of maximum wind speed data measured in the northwest Pacific hurricane records (2001-2018) from the Joint Typhoon Warning Center (Joint Typhoon

Warning Center,2018), it can be seen that the radius of maximum wind speed is inversely proportional to the central pressure difference (Fig 10). The radius of maximum wind speed has a strong relationship

with the typhoon intensity, and an empirical formula was used to calculate the radius of maximum wind speed as below:

$$R = R_0 - 0.4(P_0 - 900) + 0.01(P_0 - 900)^2 \tag{6}$$

where $P_0$ is the central air pressure (hPa), R is the radius of maximum wind speed, and $R_0$ is an empirical constant. The recommended value is 40, although this can also be adjusted by the fitting accuracy of the air pressure or the wind speed. Thus, the radius of maximum wind speed can be calculated from the central air pressure of the typhoons with five different intensities at the time of landfall, i.e., 56, 42, 38, 36, and 36 km.

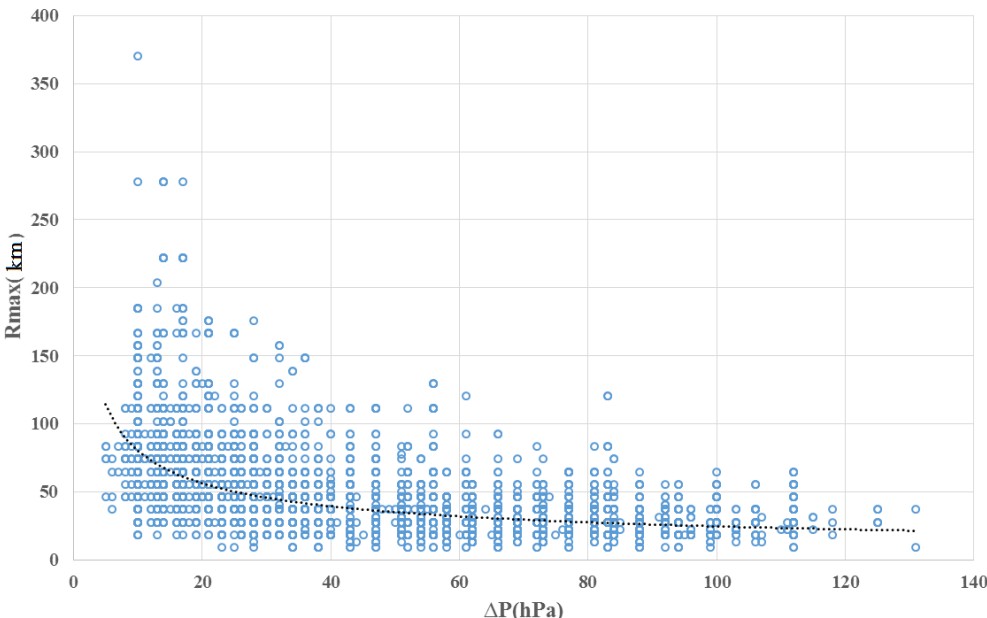

**Fig. 10 The relation between the central pressure difference ( $\Delta P$) and the radius of maximum wind ($Rmax$)**

### 3.4.4 Astronomical tide

The coupling of the astronomical tide and the storm surge was performed for simulation of the total elevation and range of inundation. The monthly averaged high tide levels of the previous 19 years during June–October at the representative tidal stations in the study area were selected as the astronomical tide levels that were coupled with the storm surge at peak surge times in the local coastal area. The astronomical tide levels at Dongtou Station on the northern side of Pingyang County and at Pipamen Station on the southern side were 2.46 and 2.33 m, respectively. The larger value of 2.46 m was taken as the astronomical tidal level of the storm surge in the storm surge numerical model for the inundation simulation.

### 3.4.5 Upstream flood runoff

The upstream flood is a factor required for numerical simulation of storm surges in estuary areas. Analysis of measured data and model studies indicate that the high-water level in an estuary area is controlled mainly by the astronomical tide and the typhoon-induced storm surge. The peak flow in an estuary has obvious influence on the high-water level during the passage of a typhoon (Sun et al., 2017). The storm surge–runoff interaction in an estuary area increases the tidal level of a typhoon-induced storm surge, resulting in a larger hazard (Zheng et al., 2013). The larger the volume of runoff is, the greater the tidal level in the estuary area will be (Hao et al. 2018). The Feiyun and Aojiang rivers, located on the northern and southern sides of Pingyang County, respectively, are the main rivers that affect the level of

flooding in Pingyang County. In this study, the superimposed upstream flood in the numerical simulation of storm surge was the average peak flows in the estuary areas of these rivers in the period of the selected historical typhoons during April-October, i.e., 1717 and 2348 m3/s for the Aojiang River and Feiyun River, respectively.

**3.4.6 Seawall collapse scenario**

The seawall is an important barrier against storm surges and excessive overtopping of waves is the main cause of seawall collapse. Overtopping waves flush the seawall or the landward slope, forming a scour pit. As the scour pit grows, the upper structure of the seawall loses support and becomes unstable (Sun et al. 2015; Zhang et al. 2017). In the design of the majority of seawalls in China's coastal areas, the wave overtopping rate, which is determined based on tide level, wave height, and seawall structure, is used as a controlling indicator and as a parameter to judge whether a seawall will collapse. According to the results of physical model tests, seawall collapse will occur when the wave overtopping rate of the coastal seawall in Pingyang County exceeds 0.05 $m^3$/s (Zhejiang Institute of Hydraulics and Estuary 2018). Once seawall collapse is determined in the numerical simulation, it will occur instantaneously without consideration of its process. After seawall collapse occurs in the numerical simulation, the ground elevation within the seawall is taken as the shoreline elevation, and the width of seawall collapse is determined by the wave overtopping rate at the representative point on the seawall. Each representative point represents a section of seawall.

**4 Calculation results**

To further analyze the accuracy of the calculation results derived from the simulations, 22 representative reference points were set along the Pingyang County coast to obtain the desired data (Fig. 11). The calculated maximum water level at each reference point for typhoons of different intensity is shown in Table 5. It can be seen that for the eastern coastal area of Pingyang County, the maximum water level of the representative points appeared near the radius of maximum wind speed on the southern side of the point of landfall of the typhoon. For the 915 and 925 hPa super typhoons, the typhoon track was moved from the reference position of Pingyang southward by 25 km, reaching the Feiyun Jiang estuary where the maximum water level of 7.78 m appeared; by 30 km, reaching the Feiyun River estuary and the eastern coast of Pingyang, where the maximum water level of 7.88 m appeared; and by 40 km, reaching the Aojiang River Estuary, where the maximum water level of 7.90 m appeared. For the 935, 945, and 965 hPa typhoons, the typhoon tracks corresponding maximum water level moved further southward as the radius of maximum wind speed increased. For example, the 965 hPa typhoon should move southward by 89 km from the reference position to reach the eastern coast of Pingyang, where the maximum water level would appear. Overall, the calculated maximum water level for the 925 hPa typhoon was approximately 0.17 to 0.53 m lower than that of the 915 hPa typhoon, but approximately 0.29 to 0.51 m higher than that of the 935 hPa typhoon. The calculated maximum water level for the 945 hPa typhoon was approximately 0.28 to0.5 m lower than that of the 935 hPa typhoon, but approximately 0.71 to 1.09 m higher than that of the 965 hPa typhoon.

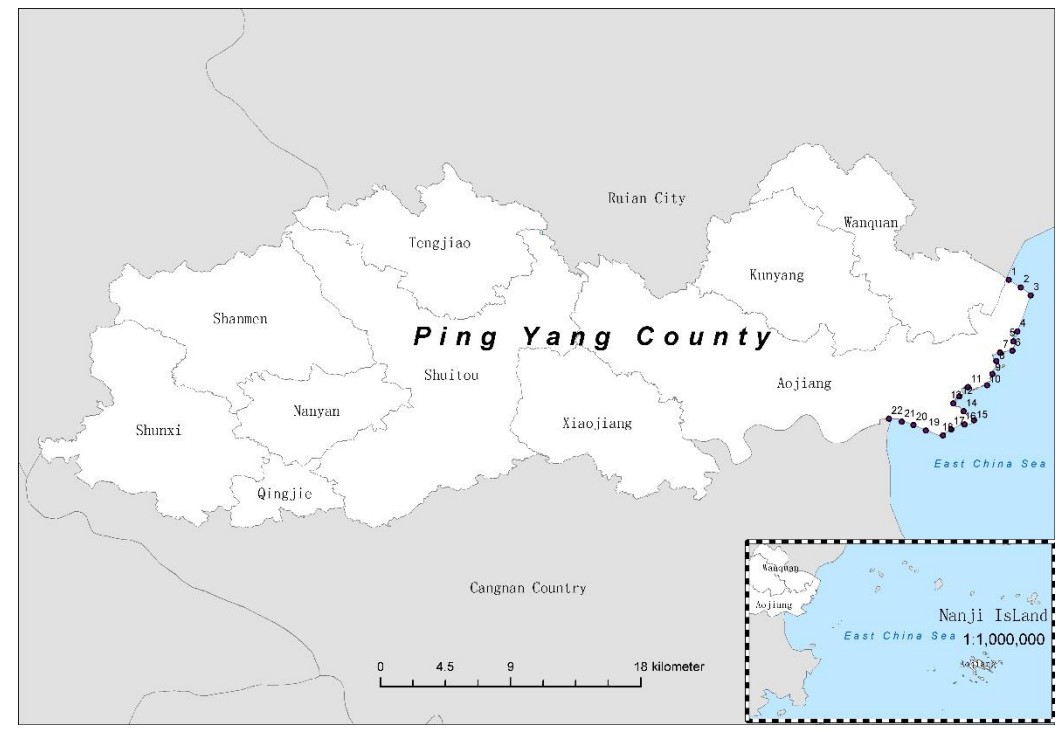

Fig.11 The reference points along Pingyang coast

Table 5 Statistical results of maximum water level associated with unfavorable tracks of typhoons under different intensity scenarios

| Reference points NO. | 915hpa | 925hpa | 935hpa | 945hpa | 965hpa |
|---|---|---|---|---|---|
| 1 | 7.61 | 7.28 | 6.90 | 6.48 | 5.53 |
| 2 | 7.59 | 7.26 | 6.87 | 6.46 | 5.51 |
| 3 | 7.60 | 7.24 | 6.85 | 6.43 | 5.48 |
| 4 | 7.62 | 7.24 | 6.83 | 6.42 | 5.46 |
| 5 | 7.65 | 7.25 | 6.82 | 6.40 | 5.44 |
| 6 | 7.70 | 7.27 | 6.82 | 6.39 | 5.42 |
| 7 | 7.79 | 7.32 | 6.86 | 6.41 | 5.42 |
| 8 | 7.85 | 7.37 | 6.89 | 6.44 | 5.44 |
| 9 | 7.87 | 7.38 | 6.90 | 6.45 | 5.44 |
| 10 | 7.88 | 7.38 | 6.90 | 6.45 | 5.44 |
| 11 | 7.88 | 7.38 | 6.91 | 6.45 | 5.45 |
| 12 | 7.87 | 7.36 | 6.90 | 6.44 | 5.44 |
| 13 | 7.91 | 7.39 | 6.94 | 6.47 | 5.46 |
| 14 | 7.58 | 7.17 | 6.72 | 6.27 | 5.37 |
| 15 | 7.66 | 7.30 | 6.84 | 6.38 | 5.40 |
| 16 | 7.66 | 7.43 | 6.95 | 6.48 | 5.45 |
| 17 | 7.79 | 7.56 | 7.06 | 6.58 | 5.52 |
| 18 | 7.90 | 7.69 | 7.18 | 6.69 | 5.60 |
| 19 | 7.81 | 7.62 | 7.15 | 6.68 | 5.68 |
| 20 | 7.64 | 7.47 | 7.05 | 6.64 | 5.72 |
| 21 | 7.44 | 7.27 | 6.91 | 6.57 | 5.75 |
| 22 | 7.36 | 7.05 | 6.76 | 6.49 | 5.77 |

Storm surge inundation was calculated for the unfavorable tracks corresponding to the five typhoon intensity scenarios. The storm tide levels caused by each unfavorable track were all the maximum storm surges of the 22 representative points around Pingyang. Therefore, it can be considered that the

inundation superposition of these unfavorable tracks represented the maximum storm surge inundation range and the water depth distribution in Pingyang County associated with the typhoons of different intensity. The range of inundation in Pingyang County by the storm surges associated with the five typhoons of different intensity is shown in Fig. 12. It can be seen that the inundation range increased with the increase of typhoon intensity. Based on Fig. 12, Table 6 shows the statistical results of the maximum inundated area corresponding to the five typhoon scenarios. It can be seen that the area of Pingyang County inundated by the storm surge associated with the 915 hPa typhoon and the most unfavourable track reached 233 km$^2$. The inundated areas included most parts of the town of Aojiang Town, eastern areas of Wanquan, north areas of Songbu, as well as parts of Kunyang and Shuitou, including administrative villages such as Qianjie and Jinmei.

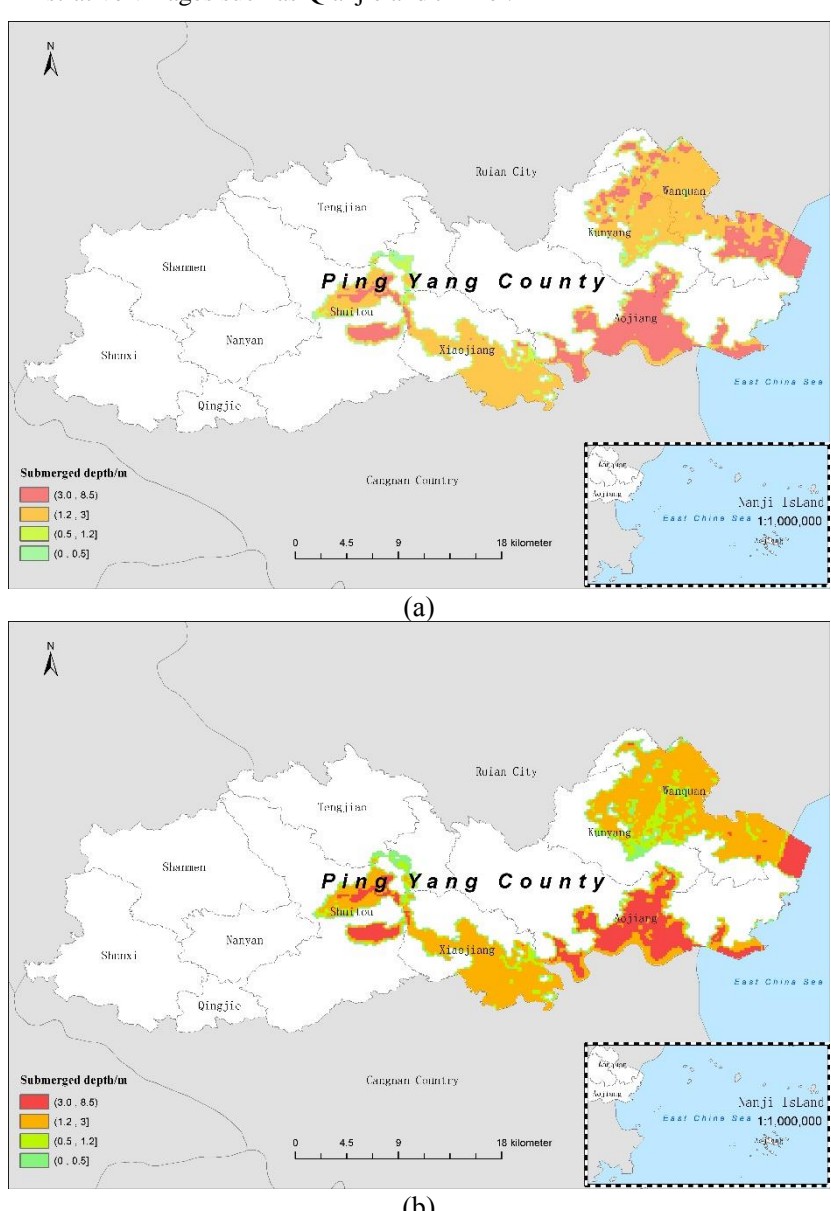

(a)

(b)

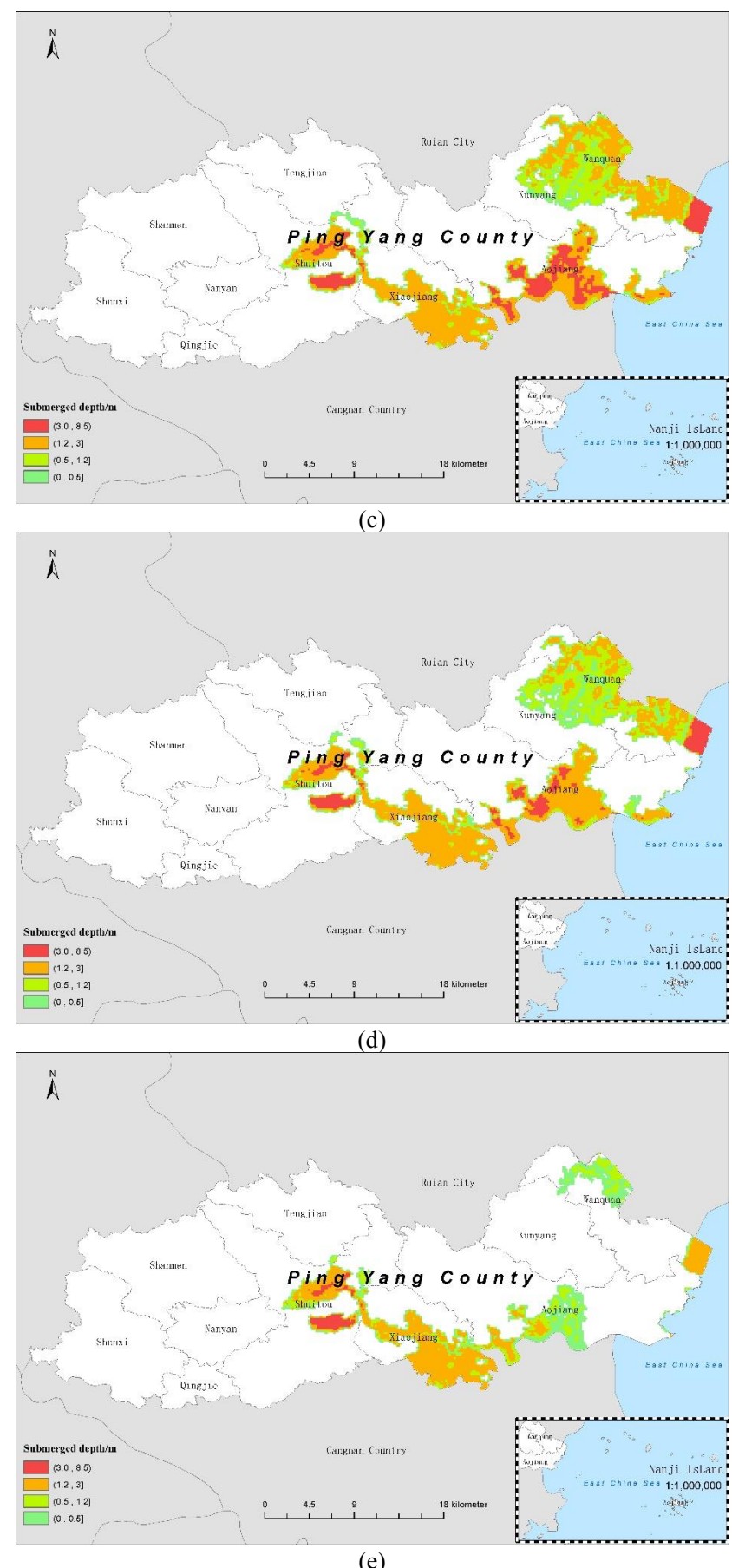

(c)

(d)

(e)

**Fig. 12 Inundation range and water depth distribution of storm surges associated with typhoons of different intensity: (a) 915 hPa, (b) 925 hPa, (c) 935 hPa, (d) 945 hPa, and (e) 965 hPa**


**Table 6 Extension of inundated areas associated with different typhoon intensity scenarios for different water level thresholds (unit: km²)**

| Typhoon Force | >3.0m | 1.2–3.0m | 0.5–1.2m | Below 0.5m | Total areas |
|---|---|---|---|---|---|
| 915hpa | 69.28 | 144.27 | 11.75 | 7.86 | 233.16 |
| 925 hpa | 44.93 | 121.10 | 22.74 | 10.97 | 199.74 |
| 935 hpa | 31.68 | 104.63 | 40.24 | 17.03 | 193.58 |
| 945 hpa | 19.68 | 97.55 | 48.33 | 22.98 | 188.54 |
| 965 hpa | 5.25 | 52.54 | 22.58 | 23.14 | 103.51 |

**5 Conclusion and discussion**

This study contributed to the methodology of storm surge inundation simulation caused by different

intensities of typhoon. A high-precision numerical model for simulating storm surges was established and validated by observational data and field-surveying inundated areas after. Using these key parameters including typhoon tracks, radius of maximum wind speed, astronomical tide, and upstream flood runoff as driving factors, the inundation extents and depths in Pingyang County corresponding to the storm surges under different typhoon intensity scenarios were simulated in combination with the storm surge

numerical model. The sea wall collapse was considered and determined by wave overtopping. Once the wave overtopping rate exceeds 0.05 m³/s, the sea wall would fail to prevent inundation caused by storm surge. The obtained results could serve as a basis for developing a methodology for storm surge disaster risk assessment in coastal areas. The study provides an insight into the spatial distribution of the areas potentially endangered by the typhoon related flooding. It can be helpful for further hazard and risk

assessments for urban planning, emergency procedures and insurance.

The inundation extent of a storm surge is related to many factors (Petroliagkis, 2018). In this study, the process of inundation is independent on the duration of the storm surge event, and a simplified sudden collapse of the seawall is assumed, which could increase the inundation range of the simulated result. The water level in the towns of Shuitou and Xiaojiang in Pingyang County is mainly caused by the

upstream flood of the Ao Jiang River. Consequently, the inundation in these two areas is directly related to upstream flood runoff. The impact of the upstream flood was only considered as the average of the flood peak flow during the storm surge in this study. The water level and inundation areas caused by the large astronomical tide due to the superposition of the extreme flood scenarios might be more unfavourable than the simulated storm surge with the superimposed average of the flood peak runoff,

which might result in uncertainty in the calculation results. We will analyze the quantitative response relationship between typhoon intensity at landfall and upstream flood runoff, and propose a quantitive method for setting flood runoff upstream of the estuary area in the further research.

This paper presents a deterministic method for setting key parameters under typhoon intensity scenarios assuming that these factors (e.g., typhoon track, radius of maximum wind speed, astronomical tide, and

upstream flood runoff) are independent. However, any correlation between these parameters is ignored. The occurrence probability of parameter combinations is difficult to evaluate. The joint probability method is an efficient way to determine the base flood elevation due to storm surge (Yang et al. 2019), and the joint probability among these factors could be established (e.g., using the Copula method) to calculate the occurrence of extreme storm surge events.

**Data availability**: All data used during the study are available from the corresponding author by request.

**Author contribution**: Shi prepared the manuscript with contributions from all co-authors and set the key parameters; Yu, Chen, Wu and Cheng performed the numerical simulation; Guo analyzed the inundated results; Sun provided the tidal observational data; Zeng conducted this research and designed the experiments.

**Competing interests**: The authors have declared that no competing interests exist.

**Acknowledgments**: This work was funded by the National Natural Science Foundation of China (41701596) and the Open-end Funds of the Key Laboratory of Coastal Disaster and Defense (Hohai University) of the Ministry of Education (201909).

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
