# Peer review of "Simulation of storm surge inundation under different typhoon intensity scenarios: Case study of Pingyang County, China"

_Natural Hazards and Earth System Sciences, 2019_

## Short Comment (SC1) · 9 Feb 2020

This study proposed a deterministic method for storm surge inundation simulation under different typhoon intensity scenarios using a numerical model. Several key parameters of typhoon activities (e.g., typhoon track, radius of maximum wind speed) as well as astronomical tide and upstream flood runoff were considered to represent the compound effect of different processes during typhoon-induced storm surge. The proposed method could provide reference for the establishment of a technical system for the assessment and zonation of storm surge risk in the coastal counties of China. Following are some suggestions for the authors which might be helpful to improve the

study:

1. What kind of data were used in this study? and the data source?

2. It would be better for the understanding the methodology if a technique flow chart could be provided.

3. It would be better if river networks and DEM could be added in the map of study area.

4. This study validated the numerical model in terms of the high tide level and the maximum storm surge at six tidal stations. However, a validation for the inundation simulation is absent, is it possible using historical flood records and marks?

5. The advantage of the proposed method should be further discussed.

---

## Referee Comment (RC1) · Anonymous Referee #1 · 6 Mar 2020

The paper aims at quantifying the inundation range and water depth distribution due to storm surges for different typhoon scenarios for the Pingyang County in China. The typhoon scenarios are constructed in a consistent way to reflect variations in tracks and intensity. The storm surges are estimated with the hydrodynamical model. In combination with the peak river runoff values the water level scenarios are used for the estimation of the coastal flooding magnitude in case of a seawall breach. The study provides an insight into the spatial distribution of the areas potentially endangered by the typhoon related flooding. It can be helpful for further hazard and risk assessments for urban planning, emergency procedures or insurance. The paper is well structured and mostly easy to follow. Here are some points requiring clarification and some sug-

gestions:

Section 3.1: Concerning the data sources and DEM, I agree with the first reviewer. Please include the response you gave, at least partially, into the paper.

Fig. 2 in the review response: Please include this figure into the paper with (1) better quality (2) color code for the land elevation, so the orography of the potentially flooded area can be deduced.

Section 3.3 and Tables 1 and 2:

l. 139 – table content is specified as "error statistics". (1) Are these values the differences between two values: observed and modeled max high water (or max storm surge) for each typhoon and location? (2) What are the "average errors" discussed in lines 142-150 and shown in the last column and line of the Tables? These values seem not to be the average (mean) of the values in the Tables. E.g. Table 1, event 9015 – Average value 8 is not equal to the mean of (3, 2, -23), similar is true for many other lines and columns. Please either specify in the text how these "average" values were obtained or correct the average values in the Tables and discussion in the Section 3.3 accordingly.

l. 142-143: the locations of the tidal stations on the map (either Fig. 1 of figure with DEM) would be helpful. They could also help to understand the significant differences in the storm surges on Fig. 4.

l. 144: "storm surge high tide" please reformulate because by definition "surge" is a residual of water level and the tide and has no tidal component.

l. 146: "10% of the maximum storm surge" – what is the value of maximum storm surge and which maximum storm surge is meant here? Is it at any particular location or/and event or averaged maximum? Please specify in the text. Also, if the 10% is about 10cm as mentioned in the text, then the maximum storm surge should be about 1m, however at the Fig. 4 there are storm surges reaching over 3m.

Sections 3.3:

The information about the tidal signal used at the open boundaries during validation is missing. Approximate tidal range at the coast is worth mentioning in this section because the discussed errors of 15-30 cm have different weight when they occur for the tidal range of e.g. 1m or 6m. Also, how the astronomical tides were estimated for calculation of storm surges is interesting, especially in connection with Fig.3 and Fig.4.

Fig. 3 and Fig. 4: Please include the dates on the x-axis additionally to the time. The axis are different between Fig. 3 and Fig. 4 and it is difficult to recognize the corresponding water level and storm surge. For example, for Ruian on Fig.4 there is a storm surge of 3m, however on Fig.3 for the same location and same typhoon it is really hard to deduce when such high surge has taken place.

l. 189: Please provide a quantitative example of the highest water level during this typhoon for any location of your choice in the area of investigation.

l. 197-198: (1) does "constant direction of movement" mean that the modified typhoon moves in a straight line? If not, please reformulate. If yes, please explain how this constant direction correspond to original typhoon track. (2) "track. . . was translated to the landing site. . ." – meaning the track was shifted so that landing points coincide? (3) the map with the original tracks of the two typhoons and the "designed typhoon" track described in the line 202 would be very helpful here

l. 215: "36 and 36 km" please correct

l. 220: change "coupled" to "linearly added" - as far as I understood, the high tide values were linearly added to the peak surge heights at the coast

l. 229: "peak flow in an estuary has no obvious influence. . ." - Does it mean the peak flow does not influence water levels during the typhoon OFF the estuary or IN the estuary? If the former, please add this to the sentence. If the later, then it contradicts with the next two sentences, where it is stated that "storm surge-runoff interaction . . .

increases the tidal level. . .".

l. 245-246: Where the wave overtopping rate came from in the numerical simulations for this study? Was the wave model additionally used to estimate the overtopping? Or how the wave overtopping was found based on the results from the storm surge model? Please specify.

Discussion: Discuss the limitations and sources of uncertainty originating in e.g. linear combination of averaged high tides instead of dynamically simulated surge and tide with their interaction; simplified seawall collapse scenarios and how this can affect the estimates of inundated areas (for example, in this study inundation is independent on the duration of the storm surge event).

---

## Short Comment (SC2) · 13 Apr 2020

The paper is overall poorly written with no scientific findings. The methods do not appear to be novel and are not sufficiently well described. It looks like a hasty paperwork without proper content in both language and techniques. No robust theory, validation, bathymetry, or topography was shown. The way of deploying local grids and river/land boundaries is probably incorrect leading to odd simulation results. Quantitative analysis is quite missing. The manuscript fails to situate the current study and results in the context of the wider literature. The presented work is not scientifically adequate for the level of an EGU journal.

---

## Referee Comment (RC2) · Anonymous Referee #2 · 3 Jun 2020

This work focuses on Simulating storm surge-induced inundation under different typhoon intensity scenarios. Although the results are within the scope of NHESS, scientific discourses on the coastal storm surge are insufficient. My suggestion is a major revision.

Comments: 1. It seems the "wave setup" is excluded in your modeling results. In my opinion, the "wave setup" is sometimes dominating the storm surge. The effect of "wave setup" is more significant than "air pressure" and even "wind stress", depended on the bathymetry. The "wave setup" effects are important to storm surge simulation and should be included in the manuscript. 2. The authors concluded that the scenario

with the most intense typhoon (915 hPa) had the most adverse track, however, many previous studies indicated that the "size" of the hurricane (typhoon) is the main factor for storm surge height and coastal inundation extent. 3. Additionally, the typhoon size is inversely proportional to the typhoon intensity if the Jelesnianski typhoon model was used. This phenomenon should be discussed in the manuscript. 4. Many previous studies revealed that using a combination of parametric typhoon model and reanalysis wind produce is more suitable for storm surge and storm wave modeling. I supposed this method is also adequate for assessing the coastal inundation.

Please also note the supplement to this comment:
https://www.nat-hazards-earth-syst-sci-discuss.net/nhess-2019-425/nhess-2019-425-RC2-supplement.pdf

---

## Author Comment (AC2) · 19 Jun 2020

The paper aims at quantifying the inundation range and water depth distribution due to storm surges for different typhoon scenarios for the Pingyang County in China. The typhoon scenarios are constructed in a consistent way to reflect variations in tracks and intensity. The storm surges are estimated with the hydrodynamical model. In combination with the peak river runoff values the water level scenarios are used for the estimation of the coastal flooding magnitude in case of a seawall breach. The study provides an insight into the spatial distribution of the areas potentially endangered by the typhoon related flooding. It can be helpful for further hazard and risk assessments

for urban planning, emergency procedures or insurance. The paper is well structured and mostly easy to follow. Here are some points requiring clarification and some suggestions: Response: We greatly appreciate your kind help in reviewing the manuscript and all constructive comments. We substantially revised the paper based on these comments.

Section 3.1: Concerning the data sources and DEM, I agree with the first reviewer. Please include the response you gave, at least partially, into the paper. Response: Thanks for your suggestion. we have added the DEM as shown in Fig 1.

Fig. 2 in the review response: Please include this figure into the paper with (1) better quality (2) color code for the land elevation, so the orography of the potentially flooded area can be deduced. Response: Thanks for your suggestion. we have modified Fig. 2b and Fig. 2c according to your advice.

Section 3.3 and Tables 1 and 2: 1ïijĽ139 – table content is specified as "error statistics". (1) Are these values the differences between two values: observed and modeled max high water (or max storm surge) for each typhoon and location? (2) What are the "average errors" discussed in lines 142-150 and shown in the last column and line of the Tables? These values seem not to be the average (mean) of the values in the Tables. E.g. Table 1, event 9015 – Average value 8 is not equal to the mean of (3, 2, -23), similar is true for many other lines and columns. Please either specify in the text how these "average" values were obtained or correct the average values in the Tables and discussion in the Section 3.3 accordingly. Response: Thanks for noting this error. we have modified the manuscript according to your advice. (1)These values are the differences between observed and modeled max high water (or max storm surge) for each typhoon and location. (2)The "average errors" indicates the mean absolute error of the differences between observed and modeled value in each station, and we have corrected the value in TABLE 1 and 2.

2ïijĽ142-143: the locations of the tidal stations on the map (either Fig. 1 of figure with

DEM) would be helpful. They could also help to understand the significant differences in the storm surges on Fig. 4. Response: Thanks for your suggestion. The locations of the tidal stations have been added in Fig 3.

3ïijL'144: "storm surge high tide" please reformulate because by definition "surge" is a residual of water level and the tide and has no tidal component. Response: Thanks for your suggestion. "storm surge high tide" has been changed to "water level" in the revised manuscript.

4ïijL'146: "10surge and which maximum storm surge is meant here? Is it at any particular location or/and event or averaged maximum? Please specify in the text. Also, if the 10about 10cm as mentioned in the text, then the maximum storm surge should be about 1m, however at the Fig. 4 there are storm surges reaching over 3m. Response: Thanks for your comments. the maximum storm surge means the max value for each tidal station in a typhoon storm surge event process. To avoid ambiguity, the sentences"10

Sections 3.3: The information about the tidal signal used at the open boundaries during validation is missing. Approximate tidal range at the coast is worth mentioning in this section because the discussed errors of 15-30 cm have different weight when they occur for the tidal range of e.g. 1m or 6m. Also, how the astronomical tides were estimated for calculation of storm surges is interesting, especially in connection with Fig.3 and Fig.4. Response: Thanks for your comments and question. The monthly averaged high tide levels of the previous 19 years during June–October at the Dongtou tidal stations in the study area were selected as the astronomical tide levels to simulate the inundation range and water depth of storm surges under different typhoon intensities in this study. The tidal range at Pingyang Coast is about 4m. In Fig.3 and Fig.4, the astronomical tides are computed based on observational water level data using harmonic analysis method during validation.

1ïijL'Fig. 3 and Fig. 4: Please include the dates on the x-axis additionally to the time.

[Figure]

The axis are different between Fig. 3 and Fig. 4 and it is difficult to recognize the corresponding water level and storm surge. For example, for Ruian on Fig.4 there is a storm surge of 3m, however on Fig.3 for the same location and same typhoon it is really hard to deduce when such high surge has taken place. Response: Thanks for your suggestion. We have modified Fig 3 and Fig 4 in the revised manuscript.

2ïïjĽ189: Please provide a quantitative example of the highest water level during this typhoon for any location of your choice in the area of investigation. Response: Thanks for your suggestion. the tidal level at Aojiang Station caused by Typhoon Fred was presented in the manuscript as below: This typhoon landed at the time of the highest astronomical tide and it generated the highest tide level ever recorded in the coastal area, causing the tidal level at Aojiang Station up to 6.56m.

3ïïjĽ197-198: (1) does "constant direction of movement" mean that the modified typhoon moves in a straight line? If not, please reformulate. If yes, please explain how this constant direction correspond to original typhoon track. (2) "track... was translated to the landing site..." – meaning the track was shifted so that landing points coincide? (3) the map with the original tracks of the two typhoons and the "designed typhoon" track described in the line 202 would be very helpful here Response: Thanks for your suggestion and comments. (1) "constant direction of movement" mean that the designed typhoon" track move in the same direction and are parallel to the track of 0608 "Saomei". (2) yes, "track... was translated tothe landing site..." means the track was shifted so that landing points coincide. (3ïïjĽthe map with the original tracks of the two typhoons and the "designed typhoon" track were presented in the revised manuscript as shown in Fig 6 and 7. 4ïïjĽ215: "36 and 36 km" please correct Response: Thanks for noting this. We have carefully checked it, and there is nothing wrong. Where the central air pressure ( $P_0$)$was set to 915 or 925 hPa, the radius of maximum wind speed (R) both obtain the value of 36 km computing by Formula 4.$

5ïïjĽ220: change "coupled" to "linearly added" - as far as I understood, the high tide values were linearly added to the peak surge heights at the coast Response: Thanks

for your suggestion. We have changed "coupled" to "linearly added" in the revised manuscript.

6ïijĽ229: "peak flow in an estuary has no obvious influence..." - Does it mean the peak flow does not influence water levels during the typhoon OFF the estuary or IN the estuary? If the former, please add this to the sentence. If the later, then it contradicts with the next two sentences, where it is stated that "storm surge-runoff interaction...increases the tidal level...". Response: Thanks for your suggestion. It means the peak flow does not influence water levels during the typhoon off the estuary after we looked up the reference (Sun et al., 2017). This sentence has been modified as below in the revised manuscript: The peak flow in an estuary has obvious influence on the high-water level during the passage of a typhoon (Sun et al., 2017).

7ïijĽ245-246: Where the wave overtopping rate came from in the numerical simulations for this study? Was the wave model additionally used to estimate the overtopping? Or how the wave overtopping was found based on the results from the storm surge model? Please specify. Response: Thanks for your question. The typhoon-astronomical-flood-wave coupled numerical model was used to perform the storm surge simulation in this study (Chen et al, 2019). Waves caused by typhoon are simulated by SWAN model, which can describe the evolution of wave fields under specific wind, flow and underwater terrain conditions in shallow waters. The governing equation is as follows. $\partial/\partial_t N + \partial/\partial_x C_x N + \partial/\partial_y C_y N + \partial/\partial\sigma$ C$\sigma$ N+$\partial/\partial\theta$ C$\theta$ N=S/$\sigma$ In the equation, N is wave action, $\sigma$ is relative frequency of waves, $\theta$ is wave direction, and S is source item. $C_x and C_y are wave propagation speed in x and y direction, respectively. C\sigma$ is propagation speed of wave action in frequency space, and C$\theta$ is the propagation speed of wave action in wave direction space. Based on the wave elements and the structural parameters of the seawall, the overtopping discharge is calculated by the empirical formula. In the simulation of dike-breaching, the varying dike top elevation is applied according to overtopping discharge to simulate the process of dike-breaching. Discussion: Discuss the limitations and sources of uncertainty originating in e.g. linear

combination of averaged high tides instead of dynamically simulated surge and tide with their interaction; simplified seawall collapse scenarios and how this can affect the estimates of inundated areas (for example, in this study inundation is independent on the duration of the storm surge event). Response: Thanks for your suggestion. According to your advice, the sections of Discussion and Conclusion has been modified in the revised manuscript. The limitation and sources of uncertainty of this study have been modified in the revised manuscript This study presents a framework of calculation of inundated areas under different typhoon intensity scenarios. The proposed framework was composed by four parts: model configuration, model validation, parameters setting and inundation simulation. Based on the historical observational data, the key parameters (e.g., typhoon track, radius of maximum wind speed, astronomical tide, and upstream flood runoff) could be set to drive the storm surge numerical model. A high-precision numerical model was established and validated for simulating storm surges within the study area. Using these key parameters as driving factors, the inundation range and water depth distribution in Pingyang County corresponding to the storm surges under different typhoon intensity scenarios were simulated in combination with the storm surge numerical model. The obtained results could serve as a basis for developing a methodology aimed at storm surge disaster risk assessment in coastal areas. The proposed method could be easily adopted in various coastal areas and serves as an effective tool for the decision making in storm surge disaster risk reduction practices. The inundation range of a storm surge is related to many factors (Petroliagkis, 2018), linear combination of averaged high tides instead of dynamically simulated surge and tide with their interaction would cause uncertainty of the simulated results. In this study, the process of inundation is independent on the duration of the storm surge event, and the seawall collapse scenarios is simplified in a sudden, which could increase the inundation range of the simulated result.The high-water level in the towns of Shuitou and Xiaojiang in Pingyang County is mainly caused by the upstream flood of the Aojiang River. Consequently, the inundation situation in these two areas is directly related to upstream flood runoff. In this study,

the impact of the upstream flood was only considered as the average of the flood peak flow during the storm surge. The water level and inundation range caused by the large astronomical tide due to the superposition of the extreme flood scenarios might be more unfavorable than the simulated storm surge with the superimposed average of the flood peak runoff, which might result in uncertainty in the calculation results. In our next study, we will further analyze the quantitative response relationship between typhoon intensity at landfall and upstream flood runoff, and propose a method for setting flood runoff upstream of the estuary area. This paper presents a deterministic method for setting key parameters under typhoon intensity scenarios assuming that these factors (e.g., typhoon track, radius of maximum wind speed, astronomical tide, and upstream flood runoff) are independent; however, any correlation between these parameters is ignored. The occurrence probability of parameter combinations is difficult to evaluate. The joint probability method is an efficient way to determine the base flood elevation due to storm surge (Yang et al. 2019), and the joint probability among these factors could be established (e.g., using the Copula method) to calculate the occurrence of extreme storm surge events. This study contributed to the methodology of quantitative assessment of storm surge hazards for coastal counties. If combined with a vulnerability curve between the loss ratio of typical exposure influenced by storm surges and the water depth induced by flooding in coastal areas, a quantitative storm surge risk could be evaluated in future research. The results of a quantitative assessment of storm surge risk could provide a theoretical basis for urban planning, development of emergency evacuation procedures, and disaster insurance.

Please also note the supplement to this comment:
https://www.nat-hazards-earth-syst-sci-discuss.net/nhess-2019-425/nhess-2019-425-AC2-supplement.pdf
* * *
Ruian City

Pingyang County

Cangnan County

DEM(m)

1900
300
0

0    5    10    20    Kilometers

N

[Figure]

**Fig. 3.** the distribution of tidal station along the coastal Zhejing Province

[Figure]

**Fig. 4.** Verification of the water level for tidal stations affected by the storm surge caused by Typhoon Fitow (No. 1323)

**Fig. 5.** Verification of the storm surge for tidal stations affected by the storm surge caused by Typhoon Fitow (No. 1323)

**Fig. 6.** Typhoon track of 9417 "Fred" and 0608"Saomei"

**Fig. 7.** "designed typhoon" track set over Pingyang County

---

## Editor Comment (EC1) · Piero Lionello (Editor) · 5 Jul 2020

The Editor thanks the reviewers for the provided comments provided.

Just a note, specifically referring to SC2. If one is raising some relevant criticalities, they need to be better explained in order to help the authors to improve their work and the editor to manage the process. Few simple unspecific sentences do not help and they are not in line with the spirit of Open Discussion of our journals

Piero Lionello

2019-425, 2020.

---

## Author Response (AR1)

Dear Editor,

Thank you very much for your consideration of our paper for the potential publication and your suggestions about the major revision. We have carefully revised the manuscript following comments point-by-point. We prepared three documents as requested: (1) a point-to-point reviewer response document including original comments/questions, our response, and corresponding revisions made in the manuscript, (2) a track-change manuscript showing all the detailed modifications in the manuscript, and (3) a clear manuscript after revision.

Again, we appreciate your kind help in the reviewing and revision process. We look forward to further updates from you.

Warm regards,
Xianwu Shi
On behalf of the co-authors

**Responses to Reference Report #1**

The paper aims at quantifying the inundation range and water depth distribution due to storm surges for different typhoon scenarios for the Pingyang County in China. The typhoon scenarios are constructed in a consistent way to reflect variations in tracks and intensity. The storm surges are estimated with the hydrodynamical model. In combination with the peak river runoff values the water level scenarios are used for the estimation of the coastal flooding magnitude in case of a seawall breach. The study provides an insight into the spatial distribution of the areas potentially endangered by the typhoon related flooding. It can be helpful for further hazard and risk assessments for urban planning, emergency procedures or insurance. The paper is well structured and mostly easy to follow. Here are some points requiring clarification and some suggestions:

**Response**: We greatly appreciate your kind help in reviewing the manuscript and all constructive comments. We substantially revised the paper based on these comments.

Section 3.1:
Concerning the data sources and DEM, I agree with the first reviewer. Please include the response you gave, at least partially, into the paper.
Response: Thanks for your suggestion. we have modified Fig 1 as bellow:

[Figure]

Fig. 1 case study area

Fig. 2 in the review response: Please include this figure into the paper with (1) better quality (2) color code for the land elevation, so the orography of the potentially flooded area can be deduced.

**Response**: Thanks for your suggestion. we have modified Fig. 2b and Fig. 2c according to your advice.

[Figure]

Fig. 2b mesh of the numerical model in the offshore area.Fig. 2c refined mesh in Pingyang County

Section 3.3 and Tables 1 and 2:

1)139 – table content is specified as "error statistics". (1) Are these values the differences between two values: observed and modeled max high water (or max storm surge) for each typhoon and location? (2) What are the "average errors" discussed in lines 142-150 and shown in the last column and line of the Tables? These values seem not to be the average (mean) of the values in the Tables. E.g. Table 1, event 9015 – Average value 8 is not equal to the mean of (3, 2, -23), similar is true for many other lines and columns. Please either specify in the text how these "average" values were obtained or correct the average values in the Tables and discussion in the Section 3.3 accordingly.

**Response**: Thanks for noting this error. we have modified the manuscript according to your advice.

(1) These values are the differences between observed and modeled max high water (or max storm surge) for each typhoon and location.

(2) The "average errors" indicates the mean absolute error of the differences between observed and modeled value in each station, and we have corrected the value in Table 1 and 2.

2)142-143: the locations of the tidal stations on the map (either Fig. 1 of figure with DEM) would be helpful. They could also help to understand the significant differences in the storm surges on Fig. 4.

**Response**: Thanks for your suggestion. A new map that the location of the tidal stations have been added was presented in the revised manuscript as below:

[Figure]

Fig.3 the distribution of tidal station along the coastal Zhejing Province

3) 144: "storm surge high tide" please reformulate because by definition "surge" is a residual of water level and the tide and has no tidal component.
**Response**: Thanks for your suggestion. "storm surge high tide" has been changed to "water level" in the revised manuscript.

4) 146: "10% of the maximum storm surge" – what is the value of maximum storm surge and which maximum storm surge is meant here? Is it at any particular location or/and event or averaged maximum? Please specify in the text. Also, if the 10% is about 10cm as mentioned in the text, then the maximum storm surge should be about 1m, however at the Fig. 4 there are storm surges reaching over 3m.
**Response**: Thanks for your comments. the maximum storm surge means the max value for each tidal station in a typhoon storm surge event process. To avoid ambiguity, the sentences"10% of the maximum storm surge" in the paper has be deleted.

Sections 3.3:
The information about the tidal signal used at the open boundaries during validation is missing. Approximate tidal range at the coast is worth mentioning in this section because the discussed errors of 15-30 cm have different weight when they occur for the tidal range of e.g. 1m or 6m. Also, how the astronomical tides were estimated for calculation of storm surges is interesting, especially in connection with Fig.3 and Fig.4.
**Response**: Thanks for your comments and question. The Global Ocean Tide Model of TPXO 6.2 developed by Oregon State University was used as tidal signal at the open boundaries. We have added the information that the tidal range along Pingyang coast is about 4.45m in the revised manuscript. The monthly averaged water levels of the previous 19 years during June–October at the Dongtou tidal stations in the study area were selected as the astronomical tide levels to simulate the inundation range and water depth of storm surges under different typhoon intensities in this study. In Fig.3 and Fig.4, the astronomical tides are computed based on observational water

level data using harmonic analysis method during validation, and we get storm surges by the simulated water level minus to the astronomical tidal value.

1)Fig. 3 and Fig. 4: Please include the dates on the x-axis additionally to the time. The axis are different between Fig. 3 and Fig. 4 and it is difficult to recognize the corresponding water level and storm surge. For example, for Ruian on Fig.4 there is a storm surge of 3m, however on Fig.3 for the same location and same typhoon it is really hard to deduce when such high surge has taken place.

**Response**: Thanks for your suggestion. We have modified Fig 3 and Fig 4 in the revised manuscript.

[Figure]

Fig. 4 Verification of the water level for tidal stations affected by the storm surge caused by Typhoon Fitow (No. 1323)

[Figure]

[Figure]

Fig. 5 Verification of the storm surge for tidal stations affected by the storm surge caused by Typhoon Fitow (No. 1323)

2)189: Please provide a quantitative example of the highest water level during this typhoon for any location of your choice in the area of investigation.

**Response**: Thanks for your suggestion. the tidal level at Ao Jiang Station caused by Typhoon Fred was presented in the manuscript as below:

This typhoon landed at the time of the highest astronomical tide and it generated the highest tide level ever recorded in the coastal area, causing the tidal level at Ao Jiang Station up to 6.56m.

3)197-198: (1) does "constant direction of movement" mean that the modified typhoon moves in a straight line? If not, please reformulate. If yes, please explain how this constant direction corresponds to original typhoon track. (2) "track… was translated to the landing site…" – meaning the track was shifted so that landing points coincide? (3) the map with the original tracks of the two typhoons and the "designed typhoon" track described in the line 202 would be very helpful here

**Response**: Thanks for your suggestion and comments.

(1) "constant direction of movement" mean that the designed typhoon" track move in the same direction and are parallel to the track of 0608 "Saomei".

(2) yes, "track… was translated to the landing site…" means the track was shifted so that landing points coincide.

(3) the map with the original tracks of the two typhoons and the "designed typhoon" track were presented in the revised manuscript as shown as bellow:

[Figure]

Fig 6 Typhoon track of 9417 "Fred" and 0608"Saomei"

[Figure]

Fig 7 "designed typhoon" track set over Pingyang County

4) 215: "36 and 36 km" please correct

**Response**: Thanks for noting this. We have carefully checked it, and there is nothing wrong.

Where the central air pressure ($P_0$) was set to 915 or 925 hPa, the radius of maximum wind speed (R) both obtain the value of 36 km computing by Formula 4.

5) 220: change "coupled" to "linearly added" - as far as I understood, the high tide values were linearly added to the peak surge heights at the coast
**Response**: Thanks for your suggestion. The coupling of the astronomical tide and the storm surge was performed for simulation of the total elevation and range of inundation, and the high tide values were not linearly added to the peak surge heights at the coast.

6) 229: "peak flow in an estuary has no obvious influence…" - Does it mean the peak flow does not influence water levels during the typhoon OFF the estuary or IN the estuary? If the former, please add this to the sentence. If the later, then it contradicts with the next two sentences, where it is stated that "storm surge-runoff interaction…increases the tidal level…".
**Response**: Thanks for your suggestion. I am sorry that there is a typo mistake in this sentence. This sentence has been modified as below in the revised manuscript:
The peak flow in an estuary has obvious influence on the high-water level during the passage of a typhoon (Sun et al., 2017).

7) 245-246: Where the wave overtopping rate came from in the numerical simulations for this study? Was the wave model additionally used to estimate the overtopping? Or how the wave overtopping was found based on the results from the storm surge model? Please specify.
**Response**: Thanks for your question. The typhoon-astronomical-flood-wave coupled numerical model was used to perform the storm surge simulation in this study (Chen et al, 2019). Waves caused by typhoon are simulated by SWAN model, which can describe the evolution of wave fields under specific wind, flow and underwater terrain conditions in shallow waters. The governing equation is as follows.

$$\frac{\partial}{\partial t}N + \frac{\partial}{\partial x}C_xN + \frac{\partial}{\partial y}C_yN + \frac{\partial}{\partial \sigma}C_\sigma N + \frac{\partial}{\partial \theta}C_\theta N = \frac{S}{\sigma}$$

In the equation, N is wave action, $\sigma$ is relative frequency of waves, $\theta$ is wave direction, and S is source item. $C_x$ and $C_y$ are wave propagation speed in $x$ and $y$ direction, respectively. $C_\sigma$ is propagation speed of wave action in frequency space, and $C_\theta$ is the propagation speed of wave action in wave direction space.

Based on the wave elements and the structural parameters of the seawall, the overtopping discharge is calculated by the empirical formula. In the simulation of dike-breaching, the varying dike top elevation is applied according to overtopping discharge to simulate the process of dike-breaching.

Discussion: Discuss the limitations and sources of uncertainty originating in e.g. linear combination of averaged high tides instead of dynamically simulated surge and tide with their interaction; simplified seawall collapse scenarios and how this can affect the estimates of inundated areas (for example, in this study inundation is independent on the duration of the storm surge event).

**Response**: Thanks for your suggestion. According to your advice, the sections of Discussion and Conclusion has been modified in the revised manuscript. The limitation and sources of uncertainty of this study have been modified in the revised manuscript as below:

*This study contributed to the methodology of storm surge inundation simulation caused by different intensities of typhoon. A high-precision numerical model for simulating storm surges was established and validated by observational data and field-surveying inundated areas after. Using these key parameters including typhoon tracks, radius of maximum wind speed, astronomical tide, and upstream flood runoff as driving factors, the inundation extents and depths in Pingyang County corresponding to the storm surges under different typhoon intensity scenarios were simulated in combination with the storm surge numerical model. The obtained results could serve as a basis for developing a methodology for storm surge disaster risk assessment in coastal areas. The study provides an insight into the spatial distribution of the areas potentially endangered by the typhoon related flooding. It can be helpful for further hazard and risk assessments for urban planning, emergency procedures and insurance.*

*The inundation extent of a storm surge is related to many factors (Petroliagkis, 2018). In this study, the process of inundation is independent on the duration of the storm surge event, and the seawall collapse scenarios is simplified in a sudden, which could increase the inundation range of the simulated result. The water level in the towns of Shuitou and Xiaojiang in Pingyang County is mainly caused by the upstream flood of the Aojiang River. Consequently, the inundation in these two areas is directly related to upstream flood runoff. The impact of the upstream flood was only considered as the average of the flood peak flow during the storm surge in this study. The water level and inundation areas caused by the large astronomical tide due to the superposition of the extreme flood scenarios might be more unfavourable than the simulated storm surge with the superimposed average of the flood peak runoff, which might result in uncertainty in the calculation results. We will analyze the quantitative response relationship between typhoon intensity at landfall and upstream flood runoff, and propose a quantitive method for setting flood runoff upstream of the estuary area in the further research.*

*This paper presents a deterministic method for setting key parameters under typhoon intensity scenarios assuming that these factors (e.g., typhoon track, radius of maximum wind speed, astronomical tide, and upstream flood runoff) are independent. However, any correlation between these parameters is ignored. The occurrence probability of parameter combinations is difficult to evaluate. The joint probability method is an efficient way to determine the base flood elevation due to storm surge (Yang et al. 2019), and the joint probability among these factors could be established (e.g., using the Copula method) to calculate the occurrence of extreme storm surge events.*

**Responses to Reference Report #2**

This work focuses on Simulating storm surge-induced inundation under different typhoon intensity scenarios. Although the results are within the scope of NHESS, scientific discourses on the coastal storm surge are insufficient. My suggestion is a major revision.

**Response**: Thanks for your comments. We substantially revised the paper based on these comments.

Comments:

1.  It seems the "wave setup" is excluded in your modeling results. In my opinion, the "wave setup" is sometimes dominating the storm surge. The effect of "wave setup" is more significant than "air pressure" and even "wind stress", depended on the bathymetry. The "wave setup" effects are important to storm surge simulation and should be included in the manuscript.

**Response**: Thanks for your suggestion. I agree with your opinion that the "wave setup" is a very important factor in storm surge simulation. "wave setup" is a phenomenon that wave breaking in nearshore cause the water level rising. In the study area of Pingyang County, the elevation of the underwater terrain at the front of the seawall is between -0.2 to -0.6m, and the slope of the front beach is slow about 1/500 to 1/1000. In the storm surge and wave simulation of typhoon intensity with 915hpa, the water depth at the front of the seawall is about 8.0~8.20m, and the corresponding effective wave height is about 4.3~4.5m, which indicate that the waves in nearshore will not be broken. In other typhoon scenarios, the waves are basically not broken. Even if some large waves break before the embankment, preliminary analysis shows that the wave setup is about 0.1~0.5m, which is less than 10% of the maximum storm surge and is still a small amount. Therefore, "wave setup" effects was not included in this study. Thanks very much for the valuable comments made by the reviewers. We will consider the impact of wave setup in the further study.

2. The authors concluded that the scenario with the most intense typhoon (915 hPa) had the most adverse track, however, many previous studies indicated that the "size" of the hurricane (typhoon) is the main factor for storm surge height and coastal inundation extent.

**Response**: Thanks for your comments. For a single storm surge event, I agree that the "size" of the typhoon is an important factor for storm surge height and coastal inundation extent. The coastal inundation caused by typhoon-induced storm surge is associated with typhoon parameters including track, intensity and typhoon size. In this study, radius of maximum wind speed which is the radius from the typhoon's center to the position where the maximum wind speed occurs was used to indicate the "size" of the typhoon, and the central pressure was used to indicate the typhoon intensity. The typhoon track was set based on the analysis of historical typhoon events who caused the most serious storm surge in Pingyang County. From the perspective of typhoon intensity, this paper use an empirical relationship between this two factors as shown in Section 4.3 to calculate the value of $Rmax$. Thus the typhoon intensity and size was set to perform the storm surge simulation in Pingyang County.

3. Additionally, the typhoon size is inversely proportional to the typhoon intensity if the Jelesnianski typhoon model was used. This phenomenon should be discussed in the manuscript.

**Response**: Thanks for your suggestion. The typhoon size has a strong connection to the typhoon intensity. As described above, radius of maximum wind speed was used to indicate the "size" of the typhoon. Collecting the historical radius of maximum wind speed data measured in the northwest

Pacific hurricane records (2001-2018) from the Joint Typhoon Warning Center (Joint Typhoon Warning Center, 2018), it can be seen that the radius of maximum wind speed is inversely proportional to the central pressure difference.

[Figure]

Fig 1 The relation between the central pressure difference ($\Delta P$) and the radius of maximum wind ($Rmax$)

In this study, the empirical relationship below was used to calculate the value of $Rmax$:

$$R = R_0 - 0.4(P_0 - 900) + 0.01(P_0 - 900)^2$$

where $P_0$ is the central air pressure (hPa), R is the radius of maximum wind speed, and $R_0$ is an empirical constant. The recommended value is 40. The $Rmax$ and typhoon intensity also presents a negative correlation in this formula.

4. Many previous studies revealed that using a combination of parametric typhoon model and reanalysis wind produce is more suitable for storm surge and storm wave modeling. I supposed this method is also adequate for assessing the coastal inundation.

**Response**: Thanks for your comments. Accurate wind forcing is an important prerequisite of storm-surge and inundation simulations.A combination of parametric typhoon model and reanalysis wind produce is more suitable for storm surge and storm wave modeling for a large scale area, parametric typhoon model is used to drive the storm surge numerical model in the area influenced by typhoon, and reanalysis wind produce is need outside of typhoon.

In this study, the track of Typhoon Saomai was selected as the designed typhoon track. The designed typhoon track was translated to a position in the middle of Pingyang County and then translated to the sides by a distance of 0.25 times the radius of maximum wind speed, until the track combination that maximized the storm surge in each coastal area of Pingyang County was determined. Pingyang County is completely located within the areas of the radius of maximum wind speed, and seriously affected by typhoon. the parametric typhoon model is enough to drive the storm surge model, and the validated results show that both the water level and storm surge obtained from the storm surge simulation are highly consistent with the actual measurements.

**Responses to Short Comment #1**

This study proposed a deterministic method for storm surge inundation simulation under different typhoon intensity scenarios using a numerical model. Several key parameters of typhoon activities (e.g., typhoon track, radius of maximum wind speed) as well as astronomical tide and upstream flood runoff were considered to represent the compound effect of different processes during typhoon-induced storm surge. The proposed method could provide reference for the establishment of a technical system for the assessment and zonation of storm surge risk in the coastal counties of China. Following are some suggestions for the authors which might be helpful to improve the study:

**Response**: Thanks for your comments. We really greatly appreciate your kind help in the reviewing the manuscript. A detailed point-by-point response was presented as below according to your comments.

1. What kind of data were used in this study? and the data source?

**Response:** Thanks for your question. Multisource data (Table 1) were collected to perform a storm surge numerical modelling in Pingyang county. The numerical model was used to simulate the storm surge inundated range and water depth. The digital elevation map (DEM) of Pingyang county and nearshore submarine topography data were collected to construct the numerical model, and tidal observational data were used to validate the model. Historical typhoon records, including time, location, and intensity, were collected to set the typhoon parameters driving the storm surge numerical model. A field survey was carried out by Zhejiang Institute of Hydraulics and Estuary to investigate the inundation situation along the Aojiang river in Pingyang County. Researchers equipped with GPS-RTK (Global Positioning Systems, Real-Time Kinematic, which supports cm-accuracy three-dimensional positioning) and rangefinders worked in two groups to make measurements from Oct.2nd to Oct.7th in 2013. The extent of the inundation was estimated based on flooding marks, such as the accumulation of trash, signs of mud, and withered plants. In addition, the range of inundation was established through interviews with local residents.

**Table 1 Multisource data used to perform storm surge numerical modelling in Pingyang County**

| Data type | Time series | Description | Source |
|---|---|---|---|
| Historical typhoon records | 1949–2018 | Time, location, and intensity of each typhoon track point | Shanghai Typhoon Institute, China Meteorological Administration |
| Digital elevation map and submarine topography | 2015 | Digital elevation map of Pingyang County and offshore submarine topography | Surveying and Mapping Bureau of Zhejiang Province |
| Tidal observational data | 1990–2015 | Hourly observational data of surge and water level for tidal station during typhoon periods | East China Sea Marine Forecasting Center, Oceanic Administration of China |
| Historical inundation ranges | 2013 | Inundation ranges caused by Fitow along the Aojiang river in Pingyang County | Field surveying by Zhejiang Institute of Hydraulics and Estuary |

2. It would be better for the understanding the methodology if a technique flow chart could be provided.

**Response**: Thanks for your suggestion. This study proposed a framework for simulation of storm surge inundation under different typhoon intensity scenarios as shown in Fig 1.

[Figure]

Fig 1 Framework of inundation simulation under different typhoon scenario in Pingyang County

3. It would be better if river networks and DEM could be added in the map of study area.

Response: Thanks for your suggestion. The river networks and DEM has been added in the map of the study area as show in Fig 2.

[Figure]

Fig 2. Case study area

4. This study validated the numerical model in terms of the high tide level and the maximum storm surge at six tidal stations. However, a validation for the inundation simulation is absent, is it possible using historical flood records and marks?

Response: Thanks for your suggestion. A validation for the inundation simulation was performed based on the inundation ranges through field surveying. The model described in section 3 of the manuscript was used to perform a simulation of the area (see Fig 3a) along the Aojiang river (Pingyang County) inundated by Typhoon Fitow. A field survey was undertaken by the Zhejiang Institute of Hydraulics and Estuary to investigate the inundation situation in Pingyang County during the storm surge disaster period caused by Fitow(see Fig 3b). The simulated and investigated inundation areas were compared (Fig. 3). It can be seen that the surveyed and simulated inundated areas are similar. The extent of the surveyed inundated area was slightly larger than that simulated because typhoon precipitation during the period of influence of Fitow caused urban waterlogging in parts of Pingyang

County.

[Figure]

(a)

(b)

Fig. 3 (a) Simulated inundated area and (b) surveyed inundated area

5. The advantage of the proposed method should be further discussed.

**Response**: Thanks for your suggestion. We have discussed the advantage of the proposed method in the revised manuscript:

*This study contributed to the methodology of storm surge inundation simulation caused by different intensities of typhoon. A high-precision numerical model for simulating storm surges was established and validated by observational data and field-surveying inundated areas after. Using these key parameters including typhoon tracks, radius of maximum wind speed, astronomical tide, and upstream flood runoff as driving factors, the inundation extents and depths in Pingyang County corresponding to the storm surges under different typhoon intensity scenarios were simulated in combination with the*

*storm surge numerical model. The obtained results could serve as a basis for developing a methodology for storm surge disaster risk assessment in coastal areas. The study provides an insight into the spatial distribution of the areas potentially endangered by the typhoon related flooding. It can be helpful for further hazard and risk assessments for urban planning, emergency procedures and insurance.*

**Responses to Short Comment #2**

The paper is overall poorly written with no scientific findings. The methods do not appear to be novel and are not sufficiently well described. It looks like a hasty paperwork without proper content in both language and techniques. No robust theory, validation, bathymetry, or topography was shown. The way of deploying local grids and river/land boundaries is probably incorrect leading to odd simulation results. Quantitative analysis is quite missing. The manuscript fails to situate the current study and results in the context of the wider literature. The presented work is not scientifically adequate for the level of an EGU journal.

Response: Thanks very much for your comments. We really greatly appreciate your kind help in the reviewing the manuscript. We substantially revised the paper based on your comments.

(1) The author team has been carefully checked the manuscript again, and some minor errors was modified in the revised manuscript. The revised manuscript has been reorganized according to two short comments and two reference report. At the same time, we asked one language professional and Editage to improve the language issues of the revised manuscript.

(2) This paper presents a method to analyze typhoon-induced storm surge under different typhoon intensities from the perspective of typhoon parameters, astronomical tide and upstream flood runoff. The proposed method was composed by four parts: model configuration, model validation, parameters setting and inundation simulation. Based on the historical observational data, the key parameters (e.g., typhoon track, radius of maximum wind speed, astronomical tide, and upstream flood runoff) could be set to drive the storm surge numerical model. The obtained results could serve as a basis for developing a methodology aimed at storm surge disaster risk assessment in coastal areas. The proposed method could be easily adopted in various coastal areas and serves as an effective tool for the decision making in storm surge disaster risk reduction practices.

(3) In the revised manuscript, the methods would be reorganized as a single section to be presented, and a technical framework will be proposed for simulation of storm surge inundation under different typhoon intensity scenarios as shown in Fig 1 with a detailed description.

(4) Fig 1 and Fig 2 were redrawn according to your advice, and the bathymetry and topography was added in the Figures. Detailed information could be found in the Response to Referent Report 1.

(5) The quantitative analysis could be found in the Section of Calculation results, and more detailed information would be presented in the revised manuscript.

Again, it will be our great honor to receive more helpful comments to improve the manuscript.

[revised manuscript text omitted]

---

## Referee Report (RR1)

The authors have well addressed all my comments, I suggest this paper can be accepted for publication in NHESS after a minor revision.

Minor comment:

1. Some references relative to the effect of typhoon winds on storm surge and wave simulation should be considered in your study (e.g., Shih-Chun Hsiao, et.al., 2020. Numerical Simulation of Large Wave Heights from Super Typhoon Nepartak (2016) in the Eastern Waters of Taiwan. J. Mar. Sci. Eng., 8, 217; Shih-Chun Hsiao, et.al., 2019. Quantifying the contribution of nonlinear interactions to storm tide simulations during a super typhoon event. Ocean Engineering, 194, 106661; Wei-Bo Chen, et. al., 2019. Wind forcing effect on hindcasting of typhoon-driven extreme waves. Ocean Engineering, 188, 106260).

---

## Editor Decision (ED1)

These are some minor suggestions, that are meant to attract more interest on your manuscript

Suggestions for a revised abstract.

China is one of the countries that are most seriously affected by storm surges. In recent years, storm surges in coastal areas of China have caused huge economic losses and a large number of human casualties. Knowledge of the inundation range and water depth of storm surges under different typhoon intensities could assist pre-disaster risk assessment and making evacuation plans, as well as provide decision support for responding to storm surges.  Taking Pingyang County in Zhejiang Province as a case study area, parameters including typhoon tracks, radius of maximum wind speed, astronomical tide, and upstream flood runoff were determined for different typhoon intensities. Numerical simulations were conducted using these parameters to investigate the inundation range and water depth distribution of storm surges in Pingyang County considering the impact of seawall collapse under five different intensity scenarios (corresponding to minimum central pressure values equal to 915, 925, 935, 945, and 965 hPa). The inundated area ranged from xxx to XXX for the most intense typhoon. The obtained results are consistent with the actual situation in the study area. The adopted procedure could be easily adopted in various coastal counties and serves as an effective tool for the decision making in storm surge disaster risk reduction practices.

- I assume that in all cases the sea wall height would have been sufficient to prevent inundation if the seawall would not have collapsed apart from the effect wave overtopping. In this correct? Could you comment in this in the conclusions? Can you add a sentence on the effect of wave overtopping without sea wall collapse?

- Table 6: improve the caption and explain in the text the meaning of "Class". I assume it refers to the water level in the inundated area, but I could not find the explanation in the manuscript. It would be useful to add a column with the total inundated area

- I read that you are not able to estimate the probability of the occurrence of typhoons as a function of their intensity. Could you anyway add some more information to put the correct perspective the thresholds that you have considered? E.g. could you add in 3.4.1 what is the intensity of the most intense recorded typhoon that affected this area? How many of them above the minimum threshold (965 hPa) that you have considered?

- Your sentence "The obtained results are consistent with the actual situation in the study area." Is not clear to me? Do you mean that the model reproduces well the observations during the Typhoon Fitow ?

- Fig.7 caption: I suggest to add "during typhoon Fitow " so that the caption would be "a) Simulated inundated area and (b) surveyed inundated area during typhoon Fitow"

- Line I suggest to replace  "and the seawall collapse scenarios is simplified in a sudden" with "and a simplified sudden collapse of the seawall is assumed"

---

## Author Response (AR2)

Dear Editor,

Thank you so much for your email. It is very glad to receive your minor revision request. We have read your comments carefully, and responded piece by piece. We hope you find the revised manuscript indeed meets the high standards of *Natural Hazards and Earth System Sciences*. Bellow please finds our detailed response to the comments.

Again, we would thank you for our kind support in the submission and revision process. We benefited a lot from the discussion and did see substantial improvement of the quality of the article.

Warm regards,
Xianwu Shi
On behalf of the co-authors

(1) Suggestions for a revised abstract.

China is one of the countries that are most seriously affected by storm surges. In recent years, storm surges in coastal areas of China have caused huge economic losses and a large number of human casualties. Knowledge of the inundation range and water depth of storm surges under different typhoon intensities could assist pre-disaster risk assessment and making evacuation plans, as well as provide decision support for responding to storm surges.  Taking Pingyang County in Zhejiang Province as a case study area, parameters including typhoon tracks, radius of maximum wind speed, astronomical tide, and upstream flood runoff were determined for different typhoon intensities. Numerical simulations were conducted using these parameters to investigate the inundation range and water depth distribution of storm surges in Pingyang County considering the impact of seawall collapse under five different intensity scenarios (corresponding to minimum central pressure values equal to 915, 925, 935, 945, and 965 hPa). The inundated area ranged from xxx to XXX for the most intense typhoon. The obtained results are consistent with the actual situation in the study area. The adopted procedure could be easily adopted in various coastal counties and serves as an effective tool for the decision making in storm surge disaster risk reduction practices.
Response: Thanks for your suggestion. We have modified the abstract according to your advice.

(2) I assume that in all cases the sea wall height would have been sufficient to prevent inundation if the seawall would not have collapsed apart from the effect wave overtopping. In this correct? Could you comment in this in the conclusions? Can you add a sentence on the effect of wave overtopping without sea wall collapse?
Response: Thanks for your question and suggestion. Seawall plays an important role in prevent inundation caused by storm surge in the coastal areas of China, and seawall collapse is mainly influenced by wave overtopping. I agree with that the sea wall height would have been sufficient to prevent inundation if the seawall would not have collapsed apart from the effect wave overtopping. A sentence was added in the Conclusion Section in the revised manuscript as below: The sea wall collapse was considered and determined by wave overtopping. Once the wave overtopping rate exceeds 0.05 $m^3$/s, the sea wall would be failed to prevent inundation caused by storm surge.

(3) Table 6: improve the caption and explain in the text the meaning of "Class". I assume it refers to the water level in the inundated area, but I could not find the explanation in the manuscript. It would be useful to add a column with the total inundated area
Response: Thanks for your question and suggestion. (1) We classified inundated area into 4 levels according to the inundated water depth, but it was not used in the paper and detailed explanation was missing in the manuscript. We have deleted "Class" in order to improve the readability of this paper. (2) A column with the total inundated area has been added in Table 6.

(4) I read that you are not able to estimate the probability of the occurrence of typhoons as a function of their intensity. Could you anyway add some more information to put the correct perspective the thresholds that you have considered? E.g. could you add in 3.4.1 what is the intensity of the most intense recorded typhoon that affected this area? How many of them above the minimum threshold (965 hPa) that you have considered?
Response: Thanks for your question and suggestion. The most intense recorded landing typhoon since 1950 that affected Pingyang County is Saomai (No. 0608), and the central pressure at the time of landing reached 920 hPa with the wind speed of 60 m/s. About 40 percent of the typhoons at the time of landing with the central air pressure were lower than 965 hPa. We have added the above information in the revised manuscript.

(5) Your sentence "The obtained results are consistent with the actual situation in the study area."
Is not clear to me? Do you mean that the model reproduces well the observations during the
Typhoon Fitow?
Response: Thanks for your question. I do mean that the model reproduces well the observations
during the Typhoon Fitow, but I suggest delete this sentence in the abstract in order to improve
the readability of this paper.

(6) Fig.7 caption: I suggest to add "during typhoon Fitow" so that the caption would be "a)
Simulated inundated area and (b) surveyed inundated area during typhoon Fitow"
Response: Thanks for your suggestion. I agree with your opinion, and "during typhoon Fitow "
has been added in the caption of Fig 7.

(7) Line I suggest to replace "and the seawall collapse scenarios is simplified in a sudden" with
"and a simplified sudden collapse of the seawall is assumed"
Response: Thanks for your suggestion. we have modified it according to your advice in the
revised manuscript.

[revised manuscript text omitted]